# Feature-aware Modulation for Learning from Temporal Tabular Data

**Hao-Run Cai**    **Han-Jia Ye**

School of Artificial Intelligence, Nanjing University, China
National Key Laboratory for Novel Software Technology, Nanjing University, China
`{caihr, yehj}@lamda.nju.edu.cn`

## Abstract

While tabular machine learning has achieved remarkable success, temporal distribution shifts pose significant challenges in real-world deployment, as the relationships between features and labels continuously evolve. Static models assume fixed mappings to ensure generalization, whereas adaptive models may overfit to transient patterns, creating a *dilemma between robustness and adaptability*. In this paper, we analyze key factors essential for constructing an effective dynamic mapping for temporal tabular data. We discover that evolving feature semantics—particularly objective and subjective meanings—introduce concept drift over time. Crucially, we identify that feature transformation strategies are able to mitigate discrepancies in feature representations across temporal stages. Motivated by these insights, we propose a **feature-aware temporal modulation** mechanism that conditions feature representations on temporal context, modulating statistical properties such as scale and skewness. By aligning feature semantics across time, our approach achieves a lightweight yet powerful adaptation, effectively balancing generalizability and adaptability. Benchmark evaluations validate the effectiveness of our method in handling temporal shifts in tabular data.

## 1  Introduction

Tabular data, structured in rows and columns, forms the foundation of critical decision-making across diverse domains such as healthcare [39], finance [42], and e-commerce [33]. Traditional machine learning models for tabular data—including tree-based methods [5, 7, 22, 36] and modern deep architectures [2, 10–13, 16–18, 52]—commonly assume independent and identically distributed (*i.i.d.*) data [3, 30]. However, in real-world scenarios [54], temporal distribution shifts [6, 41], characterized by evolving relationships between features and labels, present significant challenges. For instance, latent factors such as economic fluctuations, policy changes, or evolving user behaviors can render static models ineffective, as previously learned mappings quickly become obsolete.

Most existing approaches assume *i.i.d.* data and primarily focus on improving generalization based on a fixed mapping from the feature space to the label space. Static models, although easy to train and capable of reasonable generalization, fundamentally lack the capacity to adapt to evolving data patterns without explicit temporal awareness, as depicted in Figure 1. For example, ensemble methods [13] and classical gradient boosting decision trees (GBDTs) [7, 22, 36] exhibit **robustness** across diverse distributions [6, 41], but they lack mechanisms to capture evolving temporal dependencies.

To better understand learning under temporal shift and the limitations of existing approaches, we begin by examining the nature of evolving distribution in real-world temporal tabular datasets. Beyond the classical notions of covariate shift, label shift, and concept shift—referring respectively to changes in $p(\boldsymbol{x})$, $p(y)$, and $p(y|\boldsymbol{x})$ over time [31, 38]—temporal shifts involve additional complexities, such as dynamic transformations within the feature space $\mathcal{X}$ and the label space $\mathcal{Y}$ themselves [6]. These

39th Conference on Neural Information Processing Systems (NeurIPS 2025).

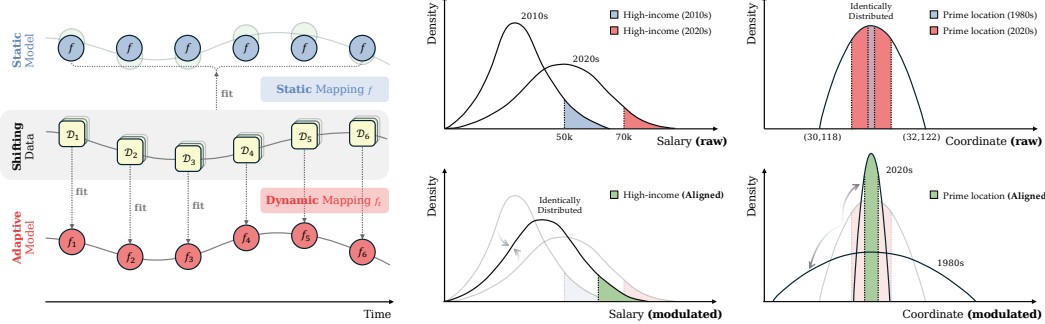

Figure 1: **Left:** Static models assume a time-invariant mapping $f$ for all temporal subset $\mathcal{D}_t$, while adaptive model $f_t$ dynamically adjust to each temporal stage. **Right:** Raw salary and location values (top) encode shifting subjective concepts like "high income" or "prime location," which vary across time. Our method aligns these semantics (bottom) by modulating feature distributions—using temporal statistics (*e.g.*, mean, std, skewness)—to preserve concept consistency over time.

challenges necessitate learning an **adaptive model** with *distributional extrapolation* capability, enabling effective generalization to previously unseen distributions.

However, learning such adaptive models is inherently difficult. Limited samples at specific time points hinder the accurate estimation of instantaneous distributions. Moreover, naïve adaptive implementations often fail to leverage temporal context effectively, limiting their ability to transfer knowledge from historical data to future deployment stages [14]. These limitations impede the development of models with *temporal extrapolation* capability, increasing the risk of overfitting to short-term patterns. This situation creates a *dilemma*: static models achieve strong generalization but ignore temporal dynamics, whereas adaptive models focus on immediate adjustments at the cost of long-term stability. Therefore, establishing a principled balance between *generalizability* and *adaptability* is essential for effectively learning from temporally evolving tabular data.

In this paper, we aim to develop an adaptive model to address temporal shifts in tabular data. We first identify a key factor in temporal generalization: the evolving *semantics* of features. Specifically, a feature's meaning can be interpreted either based on its absolute value (objective semantics) or in relation to the data distribution (subjective semantics). For example, a person's salary or a house's coordinates carry objective semantics tied to fixed numerical values. In contrast, identifying someone as part of a "high-income group" or a location as a "prime area" depends on subjective semantics, which are context-dependent and can shift over time. Thus, even if raw feature values remain unchanged, their implied meaning may evolve. Conversely, values may change while the concept remains stable. For instance, average salaries may rise over time, but the threshold for being considered "high income" is defined relative to the distribution. Similarly, coordinates remain fixed, but the concept of a "prime location" may evolve due to urban development, as illustrated in Figure 1 (right top). These evolving semantics challenge static models, which fail to interpret such changes.

Based on these insights, we propose a **feature-aware temporal modulation** mechanism that adjusts feature representations based on temporal context. We observe that the distributional statistics of features—specifically, their mean, std, and skewness—play a critical role in shaping semantics, corresponding to bias, scale, and distributional shape, as shown in Figure 2. We introduce a *learnable transformation* that modulates features according to these statistics, allowing the model to align semantics across temporal stages (Figure 1, right). This alignment helps maintain a stable relational structure among features, which is essential for learning generalizable representations under temporal shifts. As a result, the model gains distributional extrapolation capability to mitigate concept drift, and temporal extrapolation capability by leveraging similarity across time to generalize to previously unseen or sparsely observed periods. Experiments demonstrate the effectiveness of our approach in handling temporal shifts across real-world datasets. The contributions are summarized as follows:

- We analyze the challenge of temporal distribution shifts in tabular data and highlight the role of evolving feature semantics.
- We propose a novel feature-aware temporal modulation mechanism that leverages distributional statistics (mean, std, skewness) to conditionally align feature representations over time.
- Our method achieves a balance between generalization and adaptability, enabling both distributional and temporal extrapolation with low cost.

## 2 Related Work

### 2.1 Tabular Machine Learning

Tabular data is a widely-used format across various real-world applications [19–21], including healthcare [39], finance [42], and e-commerce [33]. Classical *tree-based methods*, such as Random Forest [5], XGBoost [7], LightGBM [22], and CatBoost [36], remain competitive due to their robustness, interpretability, and high performance in practice. In recent years, *deep learning approaches* for tabular data have gained significant attention. Notably, FT-Transformer (FT-T) [10] leverages the Transformer architecture to model feature interactions, while retrieval-based methods like TabR [12] and ModernNCA [52] predict labels by retrieving neighbors in the learned representation space. TabM [13] introduces an ensemble strategy that integrates multiple MLPs, while other methods, such as SNN [24], DCNv2 [50], MLP-PLR [11], and RealMLP [18], enhance the MLP architecture itself. Additionally, general-purpose approaches like TabPFN and its variants [16, 17, 26, 37] have shown remarkable versatility and generalization across various tabular tasks. Despite significant progress and strong results on established benchmarks [29, 51], the challenge of deploying these models effectively in dynamic, non-stationary environments is increasingly critical [6, 19, 41, 54].

### 2.2 Distribution Shift in Tabular Data

Existing methods for addressing distribution shift in tabular data can be broadly categorized into two groups: static methods, which aim to learn robust and generalizable representations across distributions, and adaptive methods, which adjust dynamically to distributional changes. In *static methods*, Rubachev et al. [41] observe that TabM [13] performs well under temporal shift, while Cai & Ye [6] show that GBDTs [7, 22, 36] remain competitive when using refined training protocols. *Adaptive methods*, on the other hand, focus primarily on domain shifts from a source domain to a target domain [8, 9, 23, 25, 43, 45–47]. However, these methods typically require access to target domain data in advance and are often designed to handle static domain-to-domain shifts, which are ill-suited for capturing the intra-domain dynamics common in temporal distribution shifts. Cai & Ye [6] propose incorporating temporal embedding to capture trends and periodicities, introducing temporal adaptivity into the model to some extent. In contrast, we identify key factors that contribute to model adaptability and condition feature representations on temporal embeddings through a feature-aware modulation mechanism. Our approach facilitates semantic alignment across time, overcoming the limitations of previous methods.

### 2.3 Hypernetwork

Hypernetworks, which generate the weights for another neural network [14], have become an effective strategy for making model behavior adaptive. This concept has been further developed in recent years, with specialized architectures achieving impressive results on tabular data [4, 32]. However, when applied to tabular data with temporal shifts, the conventional hypernetwork approach is considered computationally prohibitive and data-inefficient [49]. Feature-wise Linear Modulation (FiLM) approach [35] proposes a lightweight alternative, using hypernetworks for feature modulation rather than generating complete weights. FiLM has shown promising results in visual reasoning. Inspired by this, we explore effective modulation mechanisms for temporal tabular data, introducing a feature-aware temporal modulation approach that preserves the adaptive benefits of hypernetworks while ensuring computational efficiency for handling time-varying distributional shifts.

## 3 Preliminaries

### 3.1 Learning from Tabular Data with Temporal Shift

A tabular dataset with $n$ examples is generally represented as $\{(\boldsymbol{x}_i, y_i)\}_{i=1}^n$, where $\mathcal{X}$ and $\mathcal{Y}$ denote the feature space (*e.g.*, $\mathbb{R}^d$) and the label space (*e.g.*, classes or real values for classification or regression tasks), respectively. The goal is to learn a mapping $f : \mathcal{X} \to \mathcal{Y}$ from the $n$ examples, where $f(\boldsymbol{x}_i) = \hat{y}_i \approx y_i$. Typically, we assume that the pairs $(\boldsymbol{x}_i, y_i)$ are sampled *i.i.d.* from the joint distribution of $(\mathcal{X}, \mathcal{Y})$, and expect $f$ to predict the label of an unseen instance $\boldsymbol{x}^*$ sampled from the same distribution. The prediction model $f$ can be implemented using tree-based models [5, 7, 22, 36] or deep neural networks [10, 12, 13, 17, 18, 52].

In real-world applications, tabular datasets are often collected sequentially over time, meaning the underlying data distribution evolves over time. We define a *temporal* tabular dataset as $\mathcal{D} = \bigcup_t \mathcal{D}_t$, which represents a union of subsets $\mathcal{D}_t = \{(\boldsymbol{x}_i, y_i, t)\}_{i=1}^{n_t}$, where $n_t$ denotes the number of instances collected at time $t$. Each subset $\mathcal{D}_t$ is attached with a timestamp $t$. The goal remains to learn a mapping $f$ from $\bigcup_t \mathcal{D}_t$ to predict the label for a future instance $\boldsymbol{x}^*$ collected at time $t^*$. However, since the *i.i.d.* assumption no longer holds in this case, temporal shift arises both within training datasets and between test instances [6].

Moreover, for two different timestamps $t \neq t'$, there may be changes in covariant distributions (*i.e.*, $p(\boldsymbol{x}_t) \neq p(\boldsymbol{x}_{t'})$), label distributions (*i.e.*, $p(y_t) \neq p(y_{t'})$), posterior distributions (*i.e.*, $p(y_t|\boldsymbol{x}_t) \neq p(y_{t'}|\boldsymbol{x}_{t'})$) [31, 38], and even in the feature space (*i.e.*, $\mathcal{X}_t \neq \mathcal{X}_{t'}$) and label space (*i.e.*, $\mathcal{Y}_t \neq \mathcal{Y}_{t'}$) [6]. These complexities in temporal distribution shifts lead to prediction deviations in the learned mapping $f$, making it difficult to predict labels for future instances accurately.

## 3.2 Basic Solutions to Deal with Temporal Shift

We categorize methods addressing temporal shift in tabular data into two categories: static and adaptive methods. The **static method** learns a mapping $f$ that is agnostic to the time index $t$ from $\mathcal{D}$, often using empirical risk minimization [48]:

$$\min_f \sum_{(\boldsymbol{x}_i, y_i) \in \mathcal{D}} w_i \, \ell\left(y_i, \, \hat{y}_i = f(\boldsymbol{x}_i)\right) . \tag{1}$$

Here, $\ell(\cdot, \cdot)$ measures the discrepancy between the target and the predicted labels, such as cross-entropy for classification or mean squared error for regression, and $w_i$ represents the instance-specific weight to emphasize important training examples. The static mapping $f$ is trained to generalize across temporal shifts among training examples, with the expectation of being robust to future instances with varying shifts. Rubachev et al. [41] investigate the generalization ability of representative tabular models trained using this static approach.

An alternative solution is to learn a sample reweighting scheme to align the training distribution with the distributions of future instances. Various reweighting strategies have been proposed in [25, 43, 45, 46], where training sample weights are derived from test instances available during training to minimize the distribution gap between source and target domains. However, these methods assume access to target domain data during training, and are primarily designed to adapt to static domain shifts, failing to account for the dynamic and continuous nature of temporal shifts.

In contrast to fixed models, the **adaptive method** conditions the model $f$ on the timestamp, allowing the model to behave differently based on the given time. For example, [40, 44] learn adaptive weights $w_i$ in Equation (1) solely based on the training set $\mathcal{D}$, while still maintaining uniform behavior across test instances. Recently, Cai & Ye [6] proposed a temporal embedding approach, which concatenates a temporal embedding $\psi(t)$ to the model input, enabling the model to learn temporal patterns in an end-to-end manner and incorporating temporal adaptability. Specifically, $\psi(t)$ maps the timestamp $t$ to a vector, making the mapping $f(\boldsymbol{x}_i)$ become $f(\boldsymbol{x}_i, \psi(t))$, thereby incorporating temporal information. While this approach has been validated, the limited encoding dimension of the temporal embedding restricts the model's ability to extract fine-grained temporal features, and it is often affected by covariate and label shifts, resulting in fragmented feature concept alignment.

## 3.3 Challenges of the Adaptive Model for Temporal Shift

To address temporal shifts in tabular data, we aim to learn an adaptive model $f_t$ from $\mathcal{D}$. A straightforward approach is to make the parameters of $f$ dependent on the timestamp. Define an auxiliary mapping $h$ that maps the subset $\mathcal{D}_t$ to the parameters of the model $f$, and then use the generated weights for making predictions for a given input. We can express this process as $\hat{f}_t = h(\mathcal{D}_t)$. However, generating all parameters of $f$ makes the output space of $h$ large, making $h$ computationally prohibitive, data-inefficient, and prone to overfitting [4, 14, 15, 32]. To address this, we focus on a lightweight hypernetwork-based solution.

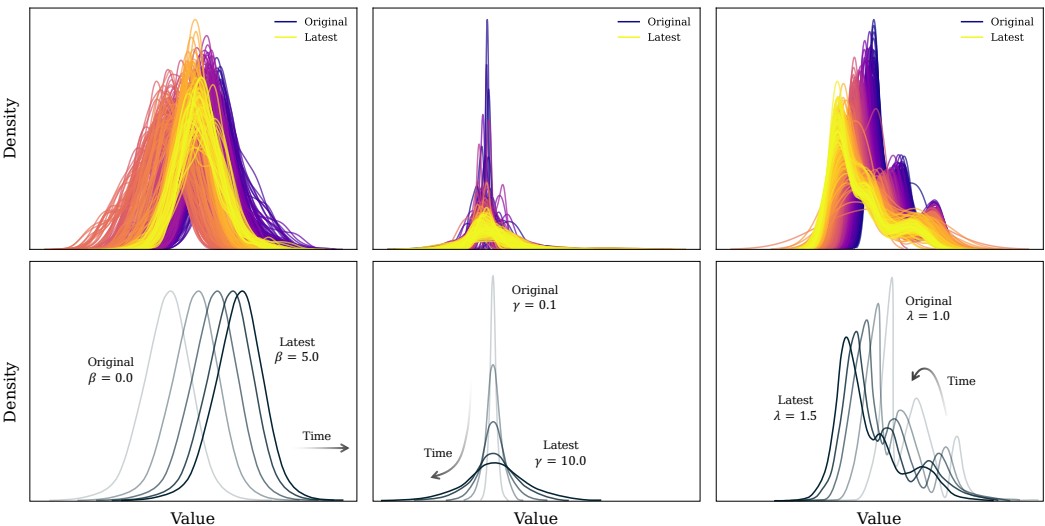

Figure 2: **Top:** Empirical feature distributions over time, with colors ranging from dark (early periods) to bright (recent periods), exhibit clear non-stationarity in bias (left), scale (middle), and skewness (right). **Bottom:** Schematic illustration of learnable transformations applied to feature distributions: shifting the mean ($\beta$) aligns bias (left), adjusting standard deviation ($\gamma$) alters scale (middle), and modulating asymmetry ($\lambda$) reshapes skewness (right). These transformations enable semantic alignment across temporal stages, thereby strengthening both generalization and adaptability.

## 4    Aligning Feature Concept under Temporal Shift

Given the evolving nature of feature semantics under temporal shifts, a central question arises: *how can we align feature representations across time to maintain semantic consistency?* Traditional approaches treat features as fixed entities, assuming their concept remain stable throughout the dataset. However, as we have observed, features with *subjective semantics*, those defined relative to the current data distribution, can shift in meaning even when their raw values remain unchanged, as discussed in Section 1 and Figure 1. This phenomenon invalidates the *i.i.d.* assumption that underpins most standard learning algorithms and motivates a feature-aware semantic modeling approach.

From a theoretical standpoint, the challenge can be viewed through the lens of *representation learning under distribution shift*. Suppose we denote the representation function as $\phi(\boldsymbol{x}; \theta)$, parameterized by $\theta$. When the semantics of $\boldsymbol{x}$ shift over time, *e.g.*, the notion of "high income" changes due to inflation, the optimal mapping $\phi$ should adapt such that semantically equivalent inputs under different $\mathcal{D}_t$ are mapped to similar representations. In other words, we seek *semantic invariance*:

$$\phi(\boldsymbol{x}; \theta_t) = \phi(\boldsymbol{x}'; \theta_{t'}), \quad \text{if } \boldsymbol{x} \sim \mathcal{D}_t, \ \boldsymbol{x}' \sim \mathcal{D}_{t'}, \text{ and } \boldsymbol{x}, \boldsymbol{x}' \text{ are semantically equivalent.}$$

This highlights the need for *temporal modeling at the feature level*, where each feature $\boldsymbol{x}_i \in \boldsymbol{x}$ may experience unique semantic shifts, and the model must dynamically adjusts its interpretation accordingly over temporal context.

Furthermore, the subjective nature of many real-world features suggests that distributional statistics, such as the *mean*, *standard deviation*, and *skewness*, can act as proxies for semantic context. Figure 2 illustrates the temporal distributions of selected features from real-world datasets within the TabReD benchmark [41]. Our analysis demonstrates that adjustments to the three aforementioned distributional statistics can effectively characterize the majority of temporal distribution shifts observed. Inspired by this, we hypothesize that semantic alignment can be achieved by *recalibrating features* informed by their temporal distributional profile. Such an approach enables the model to restore temporal invariances while embedding inductive biases conducive to extrapolation beyond observed periods.

In the following section, we operationalize this intuition by introducing a **feature-aware temporal modulation** mechanism. This mechanism conditions feature representations on temporal information through the modulation of distributional statistics computed at each timestamp. By explicitly adapting these statistical summaries, it offers a lightweight and interpretable way to align feature semantics over time, thereby improving the model's generalization under non-stationary conditions.

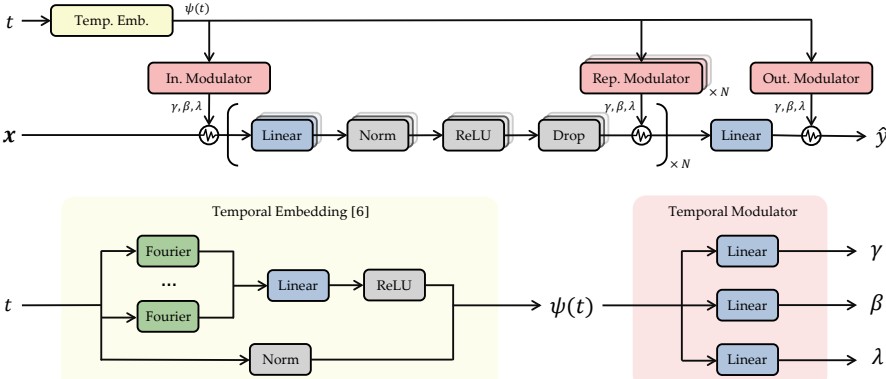

Figure 3: **Overview of our feature-aware temporal modulation framework.** Temporal modulation can be applied on raw feature input, intermediate representation, and output logits. The modulator conditions temporal context $\psi(t)$ to predict parameter $\gamma, \beta, \lambda$ for modulation based on Equation (2).

## 5 Feature-aware Temporal Modulation

To achieve semantic alignment across temporal stages, we introduce a **feature-aware temporal modulation mechanism** that conditions feature transformations on learned representations of time. The core idea is to explicitly reparameterize each input feature according to the evolving distributional context, thereby enabling the model to interpret feature values in a temporally consistent manner.

Let $t \in \mathbb{R}$ denote the timestamp associated with a given instance, and $\psi(t) \in \mathbb{R}^d$ represent the temporal embedding [6], which captures both short-term and long-term temporal dynamics. This embedding serves as a *contextual signal*, and is used as input to a lightweight modulator that outputs a set of modulation parameters for each feature dimension.

Concretely, for each feature $x \in \mathbb{R}^m$ and $t \in \mathbb{R}$, the modulator produces three parameter vectors $\gamma(\psi(t)), \beta(\psi(t)), \lambda(\psi(t)) \in \mathbb{R}^m$, which correspond to **scale**, **bias**, and a **nonlinear transformation coefficient**, respectively. We define the temporal modulation function as follows:

$$\tilde{x}_i = \gamma_i(\psi(t)) \cdot \mathrm{YJ}(x_i; \lambda_i(\psi(t))) + \beta_i(\psi(t)), \tag{2}$$

where $\mathrm{YJ}(x_i; \lambda_i)$ denotes the Yeo-Johnson transformation [53], defined as:

$$\mathrm{YJ}(x_i; \lambda_i) = \begin{cases} \left((x_i + 1)^{\lambda_i} - 1\right) / \lambda_i, & \text{if } x_i \geq 0, \\ -\left((-x_i + 1)^{2 - \lambda_i} - 1\right) / (2 - \lambda_i), & \text{if } x_i < 0. \end{cases}$$

This transformation allows the model to dynamically adjust the distribution of features by applying a smooth, nonlinear transformation that adapts to both skewed and heavy-tailed distributions. The inclusion of the Yeo-Johnson transformation enables **nonlinear reshaping** of feature distributions, which is critical for adapting to the temporal shifts in feature semantics. By leveraging the temporal context $\psi(t)$, this design provides a **feature-wise, time-dependent transformation**, where the same raw input $x_i$ can be interpreted differently depending on the temporal context $\psi(t)$, thus capturing the evolving semantics of each feature. Unlike linear modulation schemes [35], our method explicitly accommodates the nonlinear evolution of feature semantics, facilitating semantic extrapolation over time. Our temporal modulation mechanism applied to an MLP is illustrated in Figure 3. The modulation can be flexibly applied to the raw feature, within the intermediate representations, or to the predicted logits, offering enhanced adaptability across different stages of the network.

As demonstrated in the pilot study in Section 6.3 and Figure 4, the temporal adaptivity enabled by our modulation approach is grounded in a unified feature semantics. After a single modulation applied to the raw input features, the model is able to learn a stable and consistent decision boundary over the modulated representations. This explains why our method achieves superior performance without explicitly injecting temporal information into the model architecture. Moreover, while our modulation servers as an approach to handle concept drift, the covariant and label shifts inherently embedded in the temporal dimension—*i.e.*, the changes in $p(x)$ and $p(y)$ over time—are naturally mitigated through the alignment of feature semantics. Once the model learns a temporally coherent representation space, it can extract generalizable knowledge across different temporal phases, rather than overfitting to the local variations at specific timestamps.

| | Methods | HI↑ | EO↑ | HD↑ | SH↓ | CT↓ | DE↓ | MR↓ | WE↓ | **Rank** |
|---|---|---|---|---|---|---|---|---|---|---|
| | **Classical Baselines** | | | | | | | | | |
| Static Methods | Linear | 0.9388 ±0.0009 | 0.5944 ±0.0165 | 0.8231 ±0.0006 | 0.2435 ±0.0007 | 0.4864 ±0.0002 | 0.5596 ±0.0008 | 0.1680 ±0.0008 | 1.7464 ±0.0014 | 18.250 |
| | XGBoost | 0.9625 ±0.0001 | 0.6199 ±0.0005 | **0.8644** ±0.0002 | [0.2262] ±0.0002 | 0.4792 ±0.0001 | 0.5520 ±0.0000 | [0.1610] ±0.0001 | [1.4700] ±0.0011 | [6.375] |
| | CatBoost | [0.9639] ±0.0002 | 0.6242 ±0.0018 | [0.8620] ±0.0004 | 0.2292 ±0.0007 | 0.4792 ±0.0001 | 0.5495 ±0.0001 | [0.1610] ±0.0000 | [1.4654] ±0.0011 | [4.375] |
| | LightGBM | [0.9631] ±0.0001 | 0.6164 ±0.0008 | [0.8599] ±0.0006 | [0.2260] ±0.0002 | 0.4877 ±0.0002 | 0.5500 ±0.0001 | 0.1612 ±0.0000 | [1.4654] ±0.0012 | 7.375 |
| | RandomForest | 0.9580 ±0.0001 | 0.6068 ±0.0004 | 0.8171 ±0.0007 | 0.2400 ±0.0004 | 0.4841 ±0.0001 | 0.5588 ±0.0001 | 0.1647 ±0.0000 | 1.5845 ±0.0004 | 17.375 |
| | **Deep Methods** | | | | | | | | | |
| | MLP | 0.9360 ±0.0053 | 0.6220 ±0.0040 | 0.5508 ±0.0015 | 0.2641 ±0.0093 | 0.4821 ±0.0006 | 0.5515 ±0.0012 | 0.1619 ±0.0001 | 1.5362 ±0.0059 | 16.000 |
| | MLP-PLR | 0.9596 ±0.0004 | 0.6185 ±0.0072 | 0.8166 ±0.0084 | 0.2361 ±0.0026 | 0.4799 ±0.0003 | [0.5481] ±0.0011 | 0.1616 ±0.0003 | 1.5235 ±0.0038 | 10.875 |
| | FT-T | 0.9591 ±0.0048 | 0.6159 ±0.0129 | 0.5746 ±0.0319 | 0.2438 ±0.0119 | 0.4807 ±0.0008 | 0.5503 ±0.0011 | 0.1623 ±0.0003 | 1.5146 ±0.0051 | 15.125 |
| | TabR | 0.9605 ±0.0009 | 0.6148 ±0.0095 | 0.8342 ±0.0044 | 0.2370 ±0.0045 | 0.4883 ±0.0017 | 0.5550 ±0.0040 | 0.1623 ±0.0004 | 1.4732 ±0.0076 | 13.750 |
| | ModernNCA | 0.9610 ±0.0009 | [0.6341] ±0.0030 | 0.8378 ±0.0059 | 0.2325 ±0.0033 | 0.4804 ±0.0005 | 0.5520 ±0.0007 | 0.1619 ±0.0001 | 1.4857 ±0.0034 | 9.125 |
| | TabM | [0.9640] ±0.0002 | [0.6325] ±0.0042 | 0.8290 ±0.0080 | 0.2306 ±0.0030 | 0.4813 ±0.0010 | 0.5500 ±0.0014 | 0.1612 ±0.0004 | 1.4887 ±0.0049 | 7.250 |
| | **Deep Methods with Temporal Embedding [6]** | | | | | | | | | |
| Adaptive Methods | MLP | 0.9471 ±0.0038 | 0.6252 ±0.0036 | 0.5519 ±0.0013 | 0.2431 ±0.0152 | 0.4801 ±0.0004 | 0.5518 ±0.0002 | 0.1621 ±0.0002 | 1.5319 ±0.0040 | 14.375 |
| | MLP-PLR | 0.9607 ±0.0015 | 0.6110 ±0.0053 | 0.8158 ±0.0050 | 0.2338 ±0.0035 | 0.4803 ±0.0004 | 0.5494 ±0.0034 | 0.1617 ±0.0012 | 1.5133 ±0.0065 | 11.250 |
| | FT-T | 0.9608 ±0.0046 | 0.6211 ±0.0111 | 0.5563 ±0.0259 | 0.2363 ±0.0268 | [0.4778] ±0.0006 | 0.5503 ±0.0018 | 0.1622 ±0.0002 | 1.5118 ±0.0062 | 11.125 |
| | TabR | 0.9606 ±0.0038 | 0.6233 ±0.0057 | 0.8426 ±0.0140 | 0.2392 ±0.0116 | 0.4827 ±0.0043 | 0.5497 ±0.0044 | 0.1627 ±0.0003 | **1.4620** ±0.0048 | 10.125 |
| | ModernNCA | 0.9620 ±0.0004 | **0.6356** ±0.0016 | 0.8316 ±0.0088 | **0.2255** ±0.0024 | 0.4791 ±0.0005 | 0.5535 ±0.0019 | 0.1617 ±0.0002 | 1.4903 ±0.0055 | 7.625 |
| | TabM | 0.9629 ±0.0019 | 0.6271 ±0.0092 | 0.8363 ±0.0126 | 0.2321 ±0.0066 | 0.4791 ±0.0034 | [0.5488] ±0.0012 | **0.1609** ±0.0003 | 1.4812 ±0.0070 | [5.125] |
| | **Deep Methods with Temporal Modulation (Ours)** | | | | | | | | | |
| | MLP | 0.9593 ±0.0007 | 0.6230 ±0.0043 | 0.5532 ±0.0021 | 0.2345 ±0.0021 | [0.4782] ±0.0003 | 0.5502 ±0.0011 | 0.1616 ±0.0001 | 1.5179 ±0.0049 | 11.000 |
| | MLP-PLR | 0.9591 ±0.0005 | 0.6133 ±0.0054 | 0.8190 ±0.0042 | 0.2340 ±0.0027 | [0.4780] ±0.0004 | **0.5474** ±0.0008 | 0.1616 ±0.0002 | 1.5134 ±0.0045 | 10.000 |
| | TabM | **0.9641** ±0.0003 | [0.6318] ±0.0019 | [0.8457] ±0.0015 | [0.2285] ±0.0015 | **0.4773** ±0.0005 | [0.5491] ±0.0014 | [0.1610] ±0.0003 | 1.4794 ±0.0040 | **3.500** |

Table 1: **Main results on the TabReD benchmark [41].** The best performance is shown in **bold**, with the 2nd to 4th best results boxed. Metrics reported are averaged AUC↑ and RMSE↓ over 15 runs, along with the corresponding standard deviation and the average rank across all datasets. The results demonstrate that our temporal modulation approach consistently improves performance, surpassing both static models and temporal embedding baselines [6]. Our approach also enhances stability, as it naturally decouples the temporal modality from the input modality.

The ablation study in Section 6.4 and Table 2 further confirms the effectiveness of applying modulation at multiple levels of the MLP. Specifically, modulating the input features, intermediate representations, and predicted logits all contribute positively to performance. This highlights the complementary role of each modulation layer and suggests that multiple levels of temporal modulation are beneficial for capturing complex temporal dynamics.

Additionally, by sharing the temporal embedding across all instances and modulators, our method ensures both *parameter efficiency* and *temporal coherence*, enabling the model to generalize effectively across time while avoiding overfitting to specific timestamps. As we demonstrate later in our experiments, this modulation mechanism improves both predictive performance and robustness in the presence of temporal distribution shifts.

A current *limitation* of full-stage modulation lies in its incompatibility with PLR embedding [11], which is commonly integrated into state-of-the-art tabular models [11–13, 52]. PLR embedding transform numerical features into combinations of sine and cosine, resulting in an expected arcsine distribution rather than the semantically interpretable distributions discussed in Section 1 and

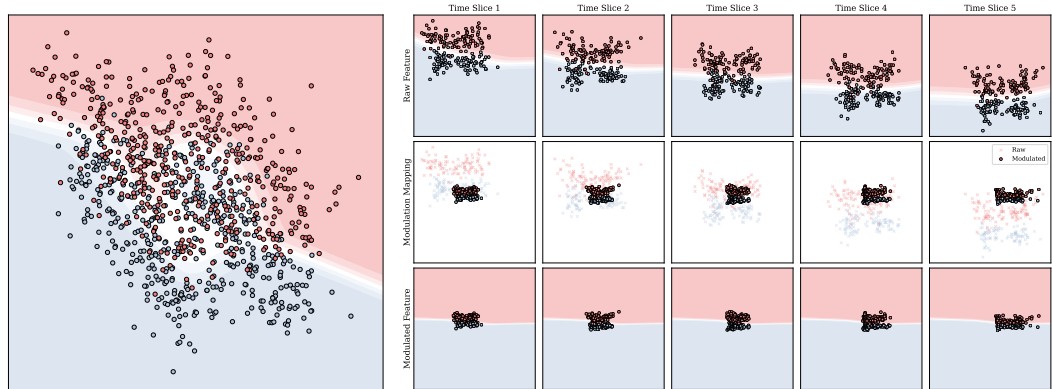

Figure 4: **Pilot study on aligning feature semantics.** The left plot illustrates the decision boundary learned by a static MLP, highlighting that such methods struggle to capture separability under temporal shifts. The top panel visualizes the evolving decision boundaries learned by an MLP with our temporal modulation, where modulation is applied once at the input layer. Each of the five subplots corresponds to a different temporal segment, revealing how the model adapts its decision boundary in response to temporal dynamics. The middle panel displays feature distributions after modulation, which are better aligned across time. This alignment enables the model to form a consistent decision boundary, as shown in the bottom panel. These results demonstrates that our lightweight modulation mechanism effectively aligns feature semantics, allowing the backbone network to operate within a unified conceptual space over time.

Section 4. This mismatch can undermine the effectiveness of our modulation strategy. Nevertheless, inspired by the ablation study, we find that applying the modulation once at the raw feature level still effective for models using PLR embedding, partially mitigating this limitation.

## 6 Experiments

### 6.1 Setup

We conduct experiments on the TabReD benchmark proposed by Rubachev et al. [41], using the refined training protocol introduced by Cai & Ye [6]. Our preprocessing, training, evaluation, and hyperparameter tuning setup follows the practices established in Cai & Ye [6]. Detailed experimental setup is provided in Section B.

### 6.2 Results

Our main results on the TabReD benchmark [41] are summarized in Table 1. Among static models, we compare classical baselines including Linear, XGBoost [7], CatBoost [36], LightGBM [22], and Random Forest [5], as well as deep tabular models such as MLP, MLP-PLR [11], FT-Transformer [10], TabR [12], ModernNCA [52], and TabM [13]. While these methods lack temporal adaptation capabilities, ensemble-based models still demonstrate strong performance due to their robustness and generalization ability. For adaptive models, we follow the approach from Cai & Ye [6] by concatenating a temporal embedding to the input features of the six aforementioned deep tabular models. This serves as a simple yet effective baseline. Most of these models benefit from this temporal input, underscoring the importance of dynamic modeling in the presence of temporal shifts.

Building on this, we apply our temporal modulation approach to MLP, MLP-PLR, and TabM. The MLP uses full modulation across all stages, while MLP-PLR and TabM employ single-step modulation on raw features, as mentioned in Section 5. Our method yields consistent performance improvements, *significantly outperforming static baselines and surpassing the temporal embedding baselines across the board*, as shown in Figure 5. Notably, although our method does not explicitly feed temporal information into the model, it still achieves robust gains, highlighting the effectiveness of our design. More specifically, we observe that even the simplest MLP achieves superior performance over most deep learning methods when equipped with full temporal modulation. *TabM with our modulation achieves the highest average rank* (3.50) among all methods. To the best of our knowledge, this is the first instance where a deep tabular method consistently outperforms GBDT-

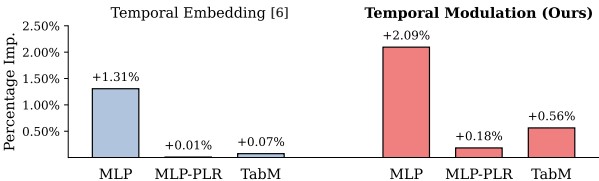

| In. | Rep. | Out. | **Imp.** (Relative) | **Rank** |
|---|---|---|---|---|
| ✗ | ✗ | ✗ | 0.00% | 7.000 |
| ✗ | ✗ | ✓ | 1.02% (48.7%) | 5.125 |
| ✗ | ✓ | ✗ | 0.26% (12.6%) | 5.250 |
| ✗ | ✓ | ✓ | 1.19% (56.8%) | 5.500 |
| ✓ | ✗ | ✗ | 1.83% (87.4%) | 3.250 |
| ✓ | ✗ | ✓ | 1.54% (73.6%) | 3.625 |
| ✓ | ✓ | ✗ | 1.62% (77.2%) | 3.750 |
| ✓ | ✓ | ✓ | **2.09%** | **2.500** |

Figure 5: **Improvement in performance with temporal modulation.** The bar chart compares the percentage improvement in performance across different models: MLP, MLP-PLR, and TabM. The left side shows results with temporal embeddings, while the right side demonstrates the improvement using our proposed temporal modulation method. Temporal modulation yields significant improvements, particularly for the MLP model, achieving a 2.09% increase, while other models exhibit more moderate gains.

Table 2: **Ablation study on the effect of modulation placement.** While all modulators have a positive impact on performance improvement, input-level modulation contributes the most to performance improvement.

based models under temporal distribution shifts. In summary, these results collectively demonstrate the strong effectiveness and generalizability of our proposed temporal modulation approach.

## 6.3 Pilot Study

The results of our pilot study, presented in Figure 4, are designed to evaluate whether our proposed modulation effectively aligns model representations. We use an MLP as the backbone and apply a single modulation on raw features to clearly separate and visualize the model's learned representations. To this end, we construct a synthetic dataset with a temporal distribution shift. Under the *i.i.d.* assumption, the dataset is non-separable, causing static models to fail in learning a valid decision boundary, as shown in Figure 4 (left). After applying our temporal modulation, the model is able to adaptively adjust its decision boundaries at each temporal stage, as illustrated in Figure 4 (right top).

We further visualize the feature distributions before and after modulation (Figure 4, right middle), and observe that the temporal shift in the input space is effectively corrected. While the post-modulation feature distributions are still non-*i.i.d.*, they become aligned enough to enable consistent and separable decision boundaries, as shown in Figure 4 (right bottom). These results demonstrate that the modulated model learns in a unified representation space and can capture temporally generalizable knowledge, thus validating our motivation of aligning feature semantics through temporal modulation.

## 6.4 Ablation Study

The results of our ablation study, presented in Table 2, investigate the relationship between model performance and the location of temporal modulation. We conduct this analysis using an MLP backbone, evaluating all combinations of the three modulation positions: raw feature input, intermediate representation, and output logits. As expected, applying modulation at all three stages yields the best performance on the TabReD benchmark, with an average improvement over the baseline MLP of 2.09%. Notably, all combinations outperform the baseline MLP without modulation, indicating that each modulation layer contributes positively to temporal generalization. These findings validate the design choice of multi-level modulation and demonstrate the complementary benefits of applying temporal adaptation throughout the network.

Beyond overall performance, we also observe several interesting findings. Applying a single modulation layer only at the raw feature level achieves 87.4% of the performance gain obtained by the full modulation setup. In contrast, removing the modulation at the raw feature layer reduces the performance gain to just 56.8% of the full modulation. This highlights the critical importance of early-stage modulation. At the raw input level, the model has access to the most complete and unaltered information, making it the most effective stage for semantic alignment. If modulation is omitted at this stage, subsequent modulations are less effective, as the model may have already internalized misaligned representations. Although any modulation applied at deeper layers of a sufficiently expressive network can approximate the optimal transformation based on the universal approximation property of neural networks in theory [34], our method achieves competitive performance with only a single modulation at the input level. This efficiency is particularly encouraging, as it enables seamless

integration of our modulation mechanism into existing models by simply adding a modulation layer at the input, without requiring changes to the internal structure of the backbone.

## 7 Conclusion

In this paper, we address temporal distribution shifts in tabular data by identifying evolving feature semantics as a core challenge. We show that conventional static and adaptive models face a trade-off between generalization and adaptability. To bridge this gap, we propose a **feature-aware temporal modulation** mechanism that conditions feature representations on temporal context through learnable transformations of distributional statistics. This enables the model to align semantics across time, facilitating both distributional and temporal extrapolation. This lightweight yet effective strategy enables stable learning under temporal shift and improves generalization to future data. Extensive experiments validate the robustness and adaptability of our method in temporal tabuar learning.

## Acknowledgments and Disclosure of Funding

This work is partially supported by Key Program of Jiangsu Science Foundation (BK20243012), CCF-ALIMAMA TECH Kangaroo Fund (NO.2024005), Collaborative Innovation Center of Novel Software Technology and Industrialization. We thank Si-Yang Liu, Yu-Sheng Li, Huai-Hong Yin, Jun-Peng Jiang, and Jin-Hui Wu for their insightful discussions. We also appreciate the reviewers for their thoughtful comments.

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

## A  Code and Data Availability

We release the complete implementation of our method at the following repository: `https://github.com/LAMDA-Tabular/Tabular-Temporal-Modulation`.

We use the TabReD [41] benchmark to evaluate the performance of our models. Furthermore, we adopt the refined training protocol and the data preprocessing procedures proposed by Cai & Ye [6].

## B  Detailed Experimental Setup

Our experiments are run under Linux using Python 3.10 and PyTorch 2.0.1. We adopt the training, evaluation, and hyperparameter tuning setups from Cai & Ye [6], Ye et al. [51], and Liu et al. [27]. Hyperparameter optimization is performed using Optuna [1], with 100 trials for most methods. Due to computational constraints, FT-Transformer and TabR are tuned with 25 trials. The search space strictly follows the configurations used in Cai & Ye [6] and Rubachev et al. [41], and is also documented in our source code (available in the `config/` folder).

Once optimal hyperparameters are identified, each method is trained using 15 random seeds, and we report the average performance across these runs. For all deep learning methods, we use a batch size of 1024 and the AdamW optimizer [28]. Classification tasks are evaluated using AUC (higher is better), while regression tasks are evaluated using RMSE (lower is better). Model selection is based on the best performance on the validation set. Following Rubachev et al. [41] and Cai & Ye [6], we adopt an early stopping strategy with a patience of 16 epochs based on validation performance.

Regarding temporal embedding, Table 3 compares the hyperparameter search spaces used in Cai & Ye [6] and in our work. While Cai & Ye [6] conducted separate hyperparameter searches for each periodic component and the trend, we adopt a unified design with a fixed embedding dimension of 128 across all periodic priors while retaining the trend component, resulting in a single hyperparameter `d_embedding` to be tuned. Unlike prior work that relies on delicate hyperparameter balancing, our formulation explicitly disentangles the input modality from the temporal modality. This separation mitigates mutual interference and alleviates the scaling issue commonly observed in temporal embeddings—where increasing dimensionality often leads to degraded performance. By contrast, our design maintains stable improvements as `d_embedding` grows, demonstrating more robust and effective modeling of temporal patterns. We further provide an ablation study in Table 4 to validate the scalability of `d_embedding`.

| Parameter | Distribution | Parameter | Distribution |
|---|---|---|---|
| `year_order` | $\{0, \mathrm{PowerInt}[1, 7]\}$ | `year_order` | Fixed 128 |
| `month_order` | $\{0, \mathrm{PowerInt}[1, 7]\}$ | `month_order` | Fixed 128 |
| `day_order` | $\{0, \mathrm{PowerInt}[1, 7]\}$ | `day_order` | Fixed 128 |
| `hour_order` | $\{0, \mathrm{PowerInt}[1, 7]\}$ | `hour_order` | Fixed 128 |
| `trend` | $\{\mathrm{True}, \mathrm{False}\}$ | `trend` | Fixed True |
| `d_embedding` | $\{0, \mathrm{PowerInt}[1, 5]\}$ | `d_embedding` | $\{0, \mathrm{PowerInt}[3, 11]\}$ |

Table 3: **Left:** Temporal embedding hyper-parameter search space in Cai & Ye [6]. **Right:** Hyper-parameter search space in this work. Here, $\mathrm{PowerInt}[a, b]$ denotes the set of integer powers of two in the range $[2^a, 2^b]$ – *e.g.*, $\mathrm{PowerInt}[1, 5] = \{2, 4, 8, 16, 32\}$.

## C  Additional Experimental Results

In this section, we include additional analyses and validations to complement the results presented in the main paper. Specifically, we provide: (1) an ablation study on the temporal embedding dimension, (2) statistical significance tests, and (3) additional pilot visualizations. Furthermore, we report the complete numerical results with standard deviations, and verify the robustness of our conclusions under alternative training protocols.

We conduct an ablation study on the temporal embedding dimension (Table 4), which shows that conventional temporal embeddings deteriorate with increasing dimensionality, whereas our modulation

| Methods | $d_{\mathrm{emb}}$ | HI↑ | EO↑ | HD↑ | SH↓ | CT↓ | DE↓ | MR↓ | WE↓ | Imp. |
|---|---|---|---|---|---|---|---|---|---|---|
| MLP (Static) | – | 0.9360 | 0.6220 | 0.5508 | 0.2641 | 0.4821 | 0.5515 | 0.1619 | 1.5362 | – |
| MLP (+Embedding) | 8 | 0.9394 | 0.6245 | 0.5515 | 0.2558 | 0.4800 | 0.5605 | 0.1621 | 1.5529 | +0.21% |
| | 32 | 0.9400 | 0.6194 | 0.5514 | 0.2497 | 0.4805 | 0.5607 | 0.1620 | 1.5448 | +0.45% |
| | 128 | 0.9442 | 0.6200 | 0.5529 | 0.2459 | 0.4799 | 0.5605 | 0.1622 | 1.5422 | +0.76% |
| | 512 | 0.9491 | 0.6260 | 0.5536 | 0.2584 | 0.4806 | 0.5591 | 0.1618 | 1.5439 | +0.40% |
| MLP (+Modulation) | 8 | 0.9439 | 0.6249 | 0.5423 | 0.2471 | 0.4786 | 0.5622 | 0.1618 | 1.5342 | **+0.65%** |
| | 32 | 0.9439 | 0.6255 | 0.5400 | 0.2507 | 0.4785 | 0.5579 | 0.1615 | 1.5317 | **+0.58%** |
| | 128 | 0.9468 | 0.6242 | 0.5547 | 0.2399 | 0.4785 | 0.5624 | 0.1617 | 1.5314 | **+1.33%** |
| | 512 | 0.9563 | 0.6213 | 0.5558 | 0.2415 | 0.4779 | 0.5589 | 0.1616 | 1.5276 | **+1.48%** |

Table 4: **Ablation on the embedding dimension** $d_{\mathrm{emb}}$. The results may be suboptimal due to the restricted hyperparameter search space. Performance of conventional temporal embeddings [6] tends to deteriorate at higher dimensions, whereas our modulation approach maintains consistent gains, demonstrating scalability and a stronger capacity to capture fine-grained temporal dependencies.

approach exhibits stable and scalable performance. The statistical significance analysis in Table 5 further confirms that our method significantly outperforms static approaches, while prior temporal embedding methods fail to achieve comparable improvements.

Additional pilot visualizations in Figure 6 analyze the behavior of our method under different types of distribution shifts. While temporal shifts are primarily concept shifts and our approach is explicitly designed to handle such cases, we observe that, in the presence of sample noise, the proposed modulation further aligns feature distributions across various shift types, enabling the backbone network to learn more stable and robust decision boundaries.

| MLP | (Static) | (+Embedding) | (+Modulation) | | TabM | (Static) | (+Embedding) | (+Modulation) |
|---|---|---|---|---|---|---|---|---|
| (Static) | 1.0000 | 0.4237 | **0.0033** | | (Static) | 1.0000 | 0.7336 | **0.0631** |
| (+Embedding) | 0.4237 | 1.0000 | 0.1122 | | (+Embedding) | 0.7336 | 1.0000 | 0.2909 |
| (+Modulation) | **0.0033** | 0.1122 | 1.0000 | | (+Modulation) | **0.0631** | 0.2909 | 1.0000 |

Table 5: **Statistical significance analysis** using Nemenyi post-hoc tests for MLP and TabM variants. The proposed modulation mechanism significantly outperforms the static baselines, whereas simply concatenating temporal embeddings [6] does not yield significant gains, demonstrating the effectiveness of our approach.

Given the absence of a standardized training protocol for temporal tabular data, we further conduct a comparative analysis of model performance under different training setups. Table 7 reports the results of each method when trained and evaluated on randomly split training and validation sets, instead of the temporal training protocol proposed by Cai & Ye [6]. In addition, Table 8 presents the results obtained under the original protocol proposed by Rubachev et al. [41]. These results show that our temporal modulation approach achieves generally competitive performance across different training protocols, highlighting its robustness to various data splitting strategies. Furthermore, we argue that the effects of temporal adaptivity and validation splitting strategies are largely orthogonal: while temporal adaptation techniques enable distributional and temporal extrapolation, the choice of validation splitting mainly determines the foundation of effective model learning.

Table 6 reports the numerical results from the ablation study discussed in the main text. We observe that the best performance on each task is consistently achieved when input-level modulation is applied, highlighting its importance for capturing temporal dynamics effectively.

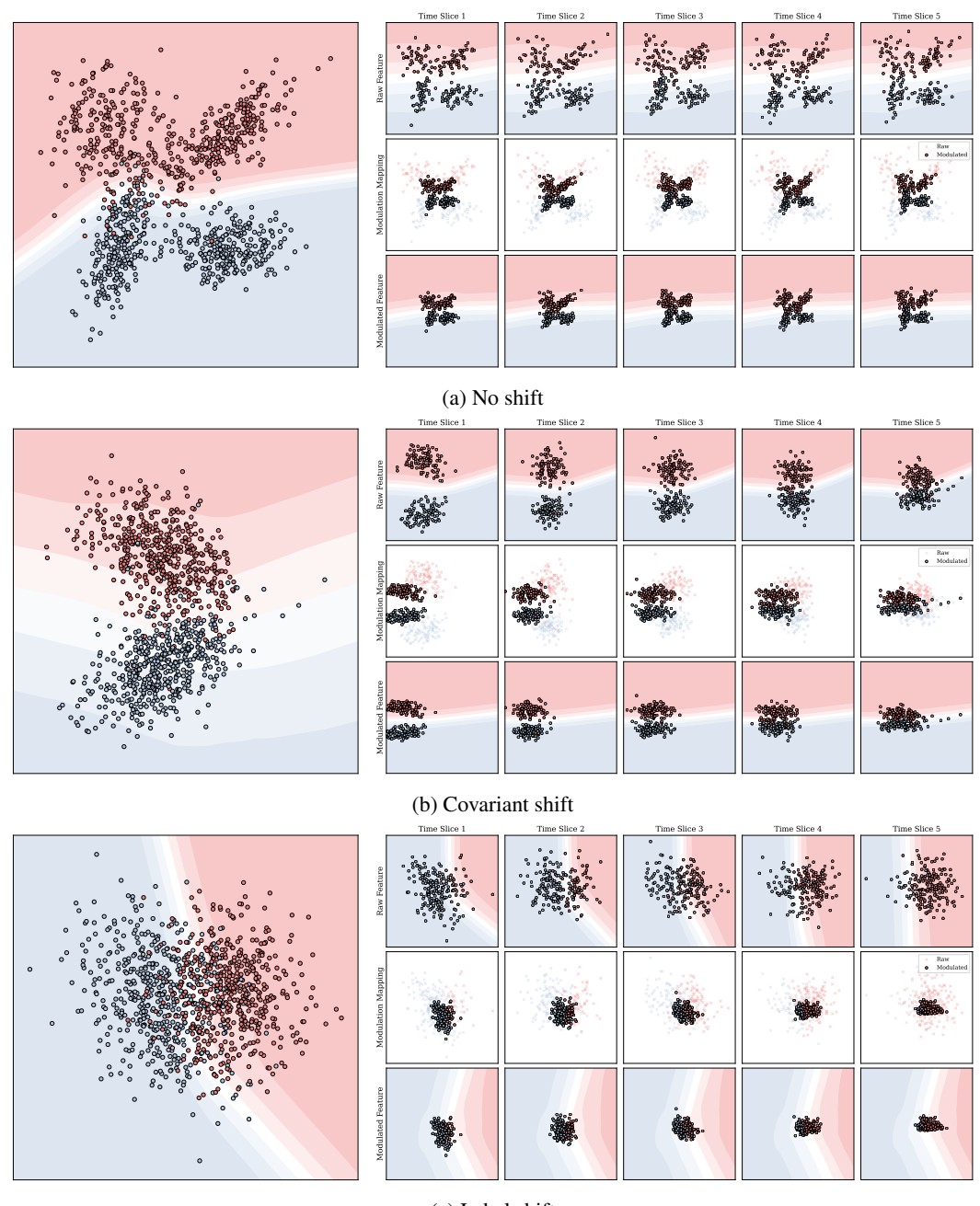

Figure 6: **Additional pilot study under different types of shifts.** While temporal shifts in tabular data predominantly manifest as concept shifts (*i.e.*, changes in $p(y|\mathbf{x})$), we further analyze our method under other types of shifts, including covariate shift (*i.e.*, changes in $p(\mathbf{x})$) and label shift (*i.e.*, changes in $p(y)$), as well as a no-shift scenario. Each subfigure follows the same visualization scheme as Figure 4, illustrating the evolving decision boundaries and feature alignments over time in our method compared to a static baseline. The results show that our temporal modulation, though primarily designed to mitigate concept shift, also promotes temporal alignment of feature distributions under covariate and label shifts, leading to more stable decision boundaries. Moreover, under the no-shift setting, it introduces no adverse effect, confirming that the modulation mechanism remains neutral when no temporal shift is present.

| In. | Rep. | Out. | HI↑ | EO↑ | HD↑ | SH↓ | CT↓ | DE↓ | MR↓ | WE↓ | **Imp.** | **Rank** |
|---|---|---|---|---|---|---|---|---|---|---|---|---|
| ✗ | ✗ | ✗ | 0.9360 | 0.6220 | 0.5508 | 0.2641 | 0.4821 | 0.5515 | 0.1619 | 1.5362 | 0.00% | 7.000 |
| ✗ | ✗ | ✓ | 0.9468 | 0.6126 | 0.5524 | 0.2460 | 0.4820 | 0.5512 | 0.1619 | 1.5176 | 1.02% | 5.125 |
| ✗ | ✓ | ✗ | 0.9400 | 0.6221 | 0.5533 | 0.2640 | 0.4814 | 0.5514 | 0.1620 | 1.5206 | 0.26% | 5.250 |
| ✗ | ✓ | ✓ | 0.9463 | 0.6185 | 0.5525 | 0.2439 | 0.4819 | 0.5508 | 0.1621 | 1.5211 | 1.19% | 5.500 |
| ✓ | ✗ | ✗ | 0.9565 | 0.6241 | 0.5519 | 0.2388 | **0.4779** | **0.5495** | 0.1623 | **1.5162** | 1.83% | 3.250 |
| ✓ | ✗ | ✓ | 0.9574 | 0.6242 | **0.5559** | 0.2392 | 0.4781 | 0.5600 | 0.1620 | 1.5355 | 1.54% | 3.625 |
| ✓ | ✓ | ✗ | 0.9567 | **0.6272** | 0.5515 | 0.2418 | 0.4782 | 0.5502 | **0.1615** | 1.5365 | 1.62% | 3.750 |
| ✓ | ✓ | ✓ | **0.9593** | 0.6230 | 0.5532 | **0.2345** | 0.4782 | 0.5502 | 0.1616 | 1.5179 | **2.09%** | **2.500** |

Table 6: **Detailed results for ablation study on the TabReD benchmark [41]** as an extension to Table 2. The best performance is shown in **bold**, with the 2nd best results underlined.

| | Methods | HI↑ | EO↑ | HD↑ | SH↓ | CT↓ | DE↓ | MR↓ | WE↓ | **Rank** |
|---|---|---|---|---|---|---|---|---|---|---|
| **Static Methods** | **Classical Baselines** | | | | | | | | | |
| | Linear | 0.9397 ±0.0009 | 0.5895 ±0.0040 | 0.8235 ±0.0018 | 0.2458 ±0.0072 | 0.4867 ±0.0003 | 0.5591 ±0.0004 | 0.1685 ±0.0073 | 1.7425 ±0.0029 | 15.125 |
| | XGBoost | 0.9625 ±0.0003 | 0.6200 ±0.0024 | 0.8452 ±0.0091 | **0.2298** ±0.0016 | 0.4806 ±0.0002 | **0.5468** ±0.0002 | 0.1611 ±0.0001 | 1.4566 ±0.0064 | 5.250 |
| | CatBoost | **0.9639** ±0.0008 | 0.6213 ±0.0025 | **0.8580** ±0.0022 | 0.2340 ±0.0017 | 0.4805 ±0.0003 | 0.5471 ±0.0002 | 0.1613 ±0.0001 | 1.4556 ±0.0021 | 5.375 |
| | LightGBM | 0.9616 ±0.0005 | 0.6136 ±0.0045 | 0.8334 ±0.0081 | 0.2322 ±0.0015 | 0.4807 ±0.0003 | 0.5469 ±0.0003 | 0.1616 ±0.0004 | **1.4471** ±0.0042 | 7.500 |
| | RandomForest | 0.9580 ±0.0002 | 0.6254 ±0.0029 | 0.8142 ±0.0017 | 0.2427 ±0.0024 | 0.4846 ±0.0001 | 0.5588 ±0.0003 | 0.1649 ±0.0000 | 1.5694 ±0.0004 | 13.250 |
| | **Deep Methods** | | | | | | | | | |
| | MLP | 0.9383 ±0.0042 | 0.6225 ±0.0031 | 0.5532 ±0.0011 | 0.2509 ±0.0119 | 0.4814 ±0.0004 | 0.5521 ±0.0006 | 0.1619 ±0.0001 | 1.5252 ±0.0059 | 13.750 |
| | MLP-PLR | 0.9599 ±0.0014 | 0.6225 ±0.0055 | 0.8208 ±0.0102 | 0.2406 ±0.0131 | 0.4800 ±0.0003 | 0.5507 ±0.0010 | 0.1616 ±0.0002 | 1.5097 ±0.0163 | 9.875 |
| | FT-T | 0.9616 ±0.0036 | 0.6268 ±0.0095 | 0.5846 ±0.0391 | 0.2369 ±0.0082 | 0.4804 ±0.0007 | 0.5503 ±0.0014 | 0.1622 ±0.0002 | 1.5001 ±0.0084 | 9.375 |
| | TabR | 0.9543 ±0.0052 | 0.6206 ±0.0055 | 0.8147 ±0.0165 | 0.2384 ±0.0068 | 0.4880 ±0.0037 | 0.5548 ±0.0033 | 0.1622 ±0.0003 | 1.4629 ±0.0067 | 12.250 |
| | ModernNCA | 0.9617 ±0.0007 | 0.6246 ±0.0110 | 0.8399 ±0.0074 | 0.2299 ±0.0037 | 0.4806 ±0.0005 | 0.5510 ±0.0015 | 0.1621 ±0.0004 | 1.4773 ±0.0101 | 7.750 |
| | TabM | 0.9629 ±0.0014 | **0.6332** ±0.0030 | 0.8282 ±0.0128 | 0.2305 ±0.0033 | 0.4794 ±0.0007 | 0.5495 ±0.0013 | **0.1607** ±0.0002 | 1.4681 ±0.0047 | 4.000 |
| **Adaptive Methods** | **Deep Methods with Temporal Embedding [6]** | | | | | | | | | |
| | MLP | 0.9451 ±0.0022 | 0.6267 ±0.0025 | 0.5530 ±0.0009 | 0.2610 ±0.0189 | 0.4796 ±0.0002 | 0.5569 ±0.0008 | 0.1618 ±0.0001 | 1.5333 ±0.0076 | 12.375 |
| | MLP-PLR | 0.9600 ±0.0009 | 0.6252 ±0.0067 | 0.8210 ±0.0079 | 0.2368 ±0.0044 | 0.4798 ±0.0020 | 0.5528 ±0.0017 | 0.1612 ±0.0001 | 1.5048 ±0.0043 | 8.625 |
| | TabM | 0.9636 ±0.0008 | 0.6299 ±0.0032 | 0.8427 ±0.0082 | 0.2317 ±0.0040 | **0.4784** ±0.0014 | 0.5492 ±0.0029 | 0.1609 ±0.0003 | 1.4891 ±0.0053 | 3.750 |
| | **Deep Methods with Temporal Modulation (Ours)** | | | | | | | | | |
| | MLP | 0.9567 ±0.0004 | 0.6267 ±0.0028 | 0.5562 ±0.0011 | 0.2499 ±0.0068 | 0.4786 ±0.0003 | 0.5594 ±0.0012 | 0.1618 ±0.0001 | 1.5122 ±0.0052 | 11.500 |
| | MLP-PLR | 0.9607 ±0.0010 | 0.6237 ±0.0041 | 0.8104 ±0.0082 | 0.2365 ±0.0027 | 0.4792 ±0.0003 | 0.5564 ±0.0010 | 0.1613 ±0.0001 | 1.5314 ±0.0078 | 9.750 |
| | TabM | 0.9633 ±0.0008 | 0.6319 ±0.0020 | 0.8380 ±0.0090 | 0.2306 ±0.0034 | 0.4785 ±0.0006 | 0.5491 ±0.0012 | 0.1608 ±0.0002 | 1.4717 ±0.0074 | **3.500** |

Table 7: **Main results on the TabReD benchmark [41] using random split,** results are averaged over three distinct partitions, with 15 seeds per partition. The best performance is shown in **bold**, with the 2nd to 4th best results boxed. Metrics reported are averaged AUC↑ and RMSE↓, along with the corresponding standard deviation and the average rank across all datasets. We follow the random splits used in Cai & Ye [6].

| Methods | | HI↑ | EO↑ | HD↑ | SH↓ | CT↓ | DE↓ | MR↓ | WE↓ | **Rank** |
|---|---|---|---|---|---|---|---|---|---|---|
| **Static Methods** | **Classical Baselines** | | | | | | | | | |
| | Linear | 0.9388 ±0.0005 | 0.5731 ±0.0058 | 0.8174 ±0.0008 | 0.2560 ±0.0134 | 0.4879 ±0.0004 | 0.5587 ±0.0008 | 0.1744 ±0.0107 | 1.7465 ±0.0031 | 14.750 |
| | XGBoost | 0.9609 ±0.0002 | 0.5764 ±0.0005 | **0.8627** ±0.0005 | 0.2475 ±0.0004 | 0.4823 ±0.0001 | **0.5459** ±0.0000 | 0.1616 ±0.0000 | **1.4699** ±0.0008 | 5.625 |
| | CatBoost | 0.9612 ±0.0003 | 0.5671 ±0.0110 | 0.8588 ±0.0005 | 0.2469 ±0.0014 | 0.4824 ±0.0001 | 0.5464 ±0.0002 | 0.1619 ±0.0000 | 1.4715 ±0.0016 | 7.000 |
| | LightGBM | 0.9600 ±0.0002 | 0.5633 ±0.0008 | 0.8580 ±0.0004 | 0.2452 ±0.0004 | 0.4826 ±0.0001 | 0.5474 ±0.0002 | 0.1618 ±0.0000 | 1.4723 ±0.0013 | 7.375 |
| | RandomForest | 0.9537 ±0.0001 | 0.5755 ±0.0008 | 0.7971 ±0.0008 | 0.2623 ±0.0006 | 0.4870 ±0.0001 | 0.5565 ±0.0001 | 0.1653 ±0.0000 | 1.5839 ±0.0004 | 14.000 |
| | **Deep Methods** | | | | | | | | | |
| | MLP | 0.9404 ±0.0026 | 0.5866 ±0.0033 | 0.4730 ±0.0006 | 0.2802 ±0.0281 | 0.4820 ±0.0005 | 0.5526 ±0.0012 | 0.1624 ±0.0001 | 1.5331 ±0.0050 | 12.125 |
| | MLP-PLR | 0.9592 ±0.0005 | 0.5816 ±0.0035 | 0.8448 ±0.0028 | 0.2412 ±0.0043 | 0.4811 ±0.0004 | 0.5533 ±0.0014 | 0.1616 ±0.0001 | 1.5185 ±0.0046 | 7.250 |
| | FT-T | 0.9562 ±0.0092 | 0.5791 ±0.0061 | 0.5301 ±0.0278 | 0.2600 ±0.0142 | 0.4814 ±0.0002 | 0.5534 ±0.0028 | 0.1627 ±0.0005 | 1.5155 ±0.0058 | 11.125 |
| | TabR | 0.9527 ±0.0024 | 0.5727 ±0.0065 | 0.8442 ±0.0041 | 0.2676 ±0.0170 | 0.4818 ±0.0003 | 0.5557 ±0.0016 | 0.1625 ±0.0005 | 1.4782 ±0.0062 | 11.625 |
| | ModernNCA | 0.9571 ±0.0060 | 0.5712 ±0.0037 | 0.8487 ±0.0018 | 0.2526 ±0.0089 | 0.4817 ±0.0007 | 0.5523 ±0.0018 | 0.1631 ±0.0001 | 1.4977 ±0.0052 | 10.000 |
| | TabM | 0.9590 ±0.0018 | 0.5952 ±0.0057 | 0.8549 ±0.0024 | 0.2465 ±0.0092 | 0.4799 ±0.0007 | 0.5522 ±0.0014 | **0.1610** ±0.0001 | 1.4852 ±0.0046 | 4.625 |
| **Adaptive Methods** | **Deep Methods with Temporal Embedding [6]** | | | | | | | | | |
| | MLP | 0.9399 ±0.0021 | 0.5877 ±0.0042 | 0.4740 ±0.0000 | 0.2795 ±0.0201 | 0.4815 ±0.0002 | 0.5589 ±0.0001 | 0.1625 ±0.0001 | 1.5363 ±0.0051 | 13.000 |
| | MLP-PLR | 0.9593 ±0.0005 | 0.5807 ±0.0035 | 0.8472 ±0.0025 | 0.2356 ±0.0022 | 0.4796 ±0.0003 | 0.5548 ±0.0027 | 0.1617 ±0.0002 | 1.5137 ±0.0042 | 6.750 |
| | TabM | 0.9616 ±0.0016 | 0.5951 ±0.0043 | 0.8548 ±0.0045 | 0.2469 ±0.0128 | **0.4795** ±0.0005 | 0.5528 ±0.0013 | 0.1615 ±0.0003 | 1.4788 ±0.0039 | 4.250 |
| | **Deep Methods with Temporal Modulation (Ours)** | | | | | | | | | |
| | MLP | 0.9538 ±0.0005 | 0.5832 ±0.0030 | 0.4753 ±0.0010 | 0.2658 ±0.0156 | 0.4811 ±0.0003 | 0.5587 ±0.0014 | 0.1623 ±0.0001 | 1.5157 ±0.0022 | 11.250 |
| | MLP-PLR | 0.9576 ±0.0010 | 0.5824 ±0.0102 | 0.8434 ±0.0029 | **0.2353** ±0.0019 | 0.4805 ±0.0004 | 0.5605 ±0.0016 | 0.1616 ±0.0001 | 1.5188 ±0.0054 | 8.625 |
| | TabM | **0.9634** ±0.0004 | **0.5978** ±0.0046 | 0.8557 ±0.0019 | 0.2415 ±0.0025 | 0.4796 ±0.0006 | 0.5533 ±0.0028 | 0.1615 ±0.0003 | 1.4813 ±0.0125 | **3.625** |

Table 8: **Main results on the TabReD benchmark [41] using original temporal splits.** The best performance is shown in **bold**, with the 2nd to 4th best results boxed. Metrics reported are averaged AUC↑ and RMSE↓ over 15 runs, along with the corresponding standard deviation and the average rank across all datasets.

# D   Computational Resources

All deep learning methods were trained on 20 NVIDIA RTX 4090 (24 GB) GPUs. Classical machine learning methods were executed on 4 Intel Xeon Platinum 8352S CPUs. All experiments were run in parallel, and the total wall-clock time for all reported experiments was approximately 14 days.

# E   Societal Impact Discussion

This work studies temporal distribution shifts in tabular data and proposes methods for improving model robustness and adaptability over time. As tabular data is widely used in high-stakes domains such as healthcare, finance, and public policy, understanding and addressing temporal shifts is critical for ensuring long-term model reliability and fairness.

Our contributions are primarily methodological and empirical. We do not use sensitive personal data, and the benchmark datasets employed in our experiments are publicly available and anonymized. While improved temporal adaptation techniques may enhance performance in real-world systems, we believe that our work supports the development of more trustworthy machine learning systems by providing tools and insights for coping with real-world temporal dynamics. We encourage future work to further investigate responsible deployment practices.

