# OpenReview forum: "Feature-aware Modulation for Learning from Temporal Tabular Data"
_NeurIPS.cc/2025/Conference — NeurIPS 2025 poster_

### Official Review · Reviewer_kWjp · 2025-06-26

**Clarity:** 3
**Significance:** 4
**Originality:** 3
**Rating:** 4
**Confidence:** 3

**Summary:**

This paper presents a method for dealing with temporal drift in tabular ML settings that uses a modulator that adjusts features and/or model outputs over time. In experiments on the TabReD benchmark they compare their approach with both static and adaptive tabular ML methods. The also include experiments on synthetic datasets, and an ablation study.

**Questions:**

1. it is not clear where and how modulation is applied in your model. I understand from Section 5 and eq. 2 that you learn mappings \psi, \gamma, \beta, and \lambda. From eq. 2 it appears that these mappings are used to modify features x before or during processing by the model. Sub-questions below.

2. How do you apply modulation? It seems the easiest way would be to modulate your input features according to eq. 2. But Figure 5 suggests other ways - e.g. during processing and after processing. Does this mean that your model must use hidden states with the same dimensionality as your input features?

3. If there are multiple ways to apply modulation, then how do you apply modulation in your experiments? I cannot determine this from your experiment description.

4. If you can apply modulation to input features prior to processing by the ML model, then you could use any ML model with this approch, right? (E.g. xgboost or SVM). In this case, why do your experiments only use neural network-based methods with modulation? It is remarkable that catboost, xgboost, and lightgbm perform so well even _without_ modulation. So why not test them _with_ modulation as well?

5. How do you learn mappings \psi, \gamma, \beta, and \lambda? This is not clear at all. Please include a description around eq. 2 and also in your experiments section for how you learned these mappings in your experiments.

**Ethical Concerns:**

["NO or VERY MINOR ethics concerns only"]

**Limitations:**

yes

**Quality:**

4

**Strengths And Weaknesses:**

This paper is very clear and well motivated. Their new method appears both novel and technically sound (see below for some questions and concerns). Their experiments appear to be very robust, and they clearly demonstrate the utility of their modulation approach. I have several questions and concerns about their method described in the questions section.

---

> ### Author Rebuttal · Authors · 2025-07-31
>
> Thank you for your valuable feedback! We will address your concerns in the following responses.
>
> > How do you apply modulation?
>
> Our feature-aware temporal modulation can be **flexibly applied at three stages**: input features, intermediate representations, or output logits (**L226**). The ablation study in **Table 2** explores the relationship between the application position of modulation and performance improvement, and it is found that modulation at all stages has a positive impact on performance (**Sec. 6.4**).
>
> > Does this mean that your model must use hidden states with the same dimensionality as your input features?
>
> This design does not require hidden states to match input dimensionality. Modulation parameters are dynamically generated per layer, allowing dimension-agnostic adaptation.
>
> > How do you apply modulation in your experiments?
>
> As specified in Sec. 6.2 (**L274**), the MLP uses full modulation across all stages, while MLP-PLR and TabM only employ single-step modulation on input features.
>
> > How do you learn $\psi$, $\gamma$, $\beta$, and $\lambda$?
>
> All parameters of the temporal embedding and modulator are **jointly optimized with the backbone model** using the AdamW optimizer and standard losses (e.g., cross-entropy for classification and RMSE for regression).
>
> > You could use any ML model with this approch, right? Why do your experiments only use neural network-based methods with modulation?
>
> **The modulation mechanism is not decoupled from model training**. Instead, it is learned through end-to-end optimization and relies on gradient-based methods.
>
> Consequently, our modulation strategy is compatible with any deep learning model, but non-differentiable models (e.g., tree-based ensembles) are unable to optimize the parameters of $\psi$, $\gamma$, $\beta$, and $\lambda$ due to the absence of gradient flow.
>
> **In summary**, our modulation is a deep learning-specific, gradient-based adaptation mechanism applied flexibly across network stages. Its strength lies in jointly learning temporal representations and feature transformations, which is infeasible for non-differentiable models. We appreciate your comment and will acknowledge this limitation in our revision.
>
> We hope this response addresses your concern. Please feel free to raise any further questions or suggestions!

---

> ### Author Response · Authors · 2025-08-06
>
> Dear Reviewer,
>
> I hope this message finds you well. As the discussion period is nearing its end, with about three days remaining, I wanted to ensure that we have addressed all your concerns satisfactorily. If there are any additional points or feedback you would like us to consider, please let us know. Your insights are invaluable to us, and we are eager to address any remaining issues to improve our work.
>
> Thank you for your time and effort in reviewing our paper.

---

> ### Author Response · Authors · 2025-08-09
>
> Dear Reviewer,
>
> We greatly appreciate your valuable feedback. With less than 10 hours remaining in the rebuttal phase, we sincerely hope our previous responses have addressed your concerns. If you have any further questions, we would be more than happy to clarify them within the remaining time. We look forward to hearing from you.

---

### Official Review · Reviewer_UGg7 · 2025-07-02

**Clarity:** 3
**Significance:** 3
**Originality:** 3
**Rating:** 5
**Confidence:** 5

**Summary:**

This work is motivated by the challenges of modeling tabular data that shifts over time. They propose an efficient feature-aware temporal modulation mechanism, inspired by feature-wise affine transformations such as FiLM, that adjusts feature representations based on temporal context in a learnable way, thereby allowing for the modulation of features over time. The proposed method is evaluated on a series of distributional and temporal extrapolation tasks from the TabRed benchmark.

The method is intuitively motivated — semantics of features vary over time, and can be modulated based on “subjective semantics” (i.e. with reference to the rest of the data distribution at a similar point in time), and the results are fairly clearly in favor of the proposed method, at least as applied to the TabRed benchmark. I raise a few questions below, but feel there is a strong case for acceptance for this paper.

**Questions:**

See above.

**Ethical Concerns:**

["NO or VERY MINOR ethics concerns only"]

**Final Justification:**

I have read the author response, and will keep my score of 5. As I said in my original review, I feel there is a clear case for acceptance for this work.

**Limitations:**

See above.

**Paper Formatting Concerns:**

None.

**Quality:**

4

**Strengths And Weaknesses:**

# Major comments

* The paper is well-written and does a good job setting up some of the main issues in this field, for example, the tension between generalizability and adaptability for temporal models.
* The TabRed benchmark is certainly one that aligns with the authors’ intended domain application, but it is also not very widely used. At the same time, many other tabular benchmarks contain at least a subset of tables with temporal columns. It would be nice to see these results applied to more “standard” tabular benchmarks (but where there is temporal information that is often not fully utilized) — this would provide further evidence of the advantages of the proposed approach, and would particularly emphasize how utilizing temporal information “as temporal information” would improve the model performance.
* In principle, it seems that there is nothing limiting this approach to temporal shifts — indeed, if this improves performance by modulating feature values over time, it would also be interesting to see whether it improves performance by similarly modulating based on the value of another continuous (or categorical) feature, effectively swapping the “time” modulator for some other signal. (This is a direction for future work, but an interesting one that would require effectively no changes to the authors’ methods.)

# Minor comments
* L258: “Our preprocessing, training, evaluation, and hyperparameter tuning setup follows the practices established in Cai & Ye [5]” - please provide some further details here.
* I find the colors of the points in Figure 4 quite difficult to see, especially due to the small size. Perhaps this can be improved with (1) a higher-contrast palette and (2) changing the marker shape for different classes. I also feel that the figure would be easier to interpret with some indication of the values of the “time axis” for the smaller panels in the plot (perhaps by labeling the title of each subfigure with the time range it represents).

# Typos etc
* L249 “These embedding transform” —> these embeddings
* L261 “Unlike prior work [5] requires” —> work [5], which requires

---

> ### Author Rebuttal · Authors · 2025-07-31
>
> Thank you for your thorough review and insightful feedback! We will address your concerns in the following responses.
>
> > It would be nice to see these results applied to more “standard” tabular benchmarks.
>
> We acknowledge that TabReD has not received the attention it deserves. However, even if the datasets in existing benchmarks contain temporal columns, they are often not suitable for direct evaluation. These problems include data leakage caused by non-temporal splitting, extensive feature engineering of timestamps, and sample grouping that is not suitable for temporal splitting. **This has limited our options.** The Appendix of the TabReD paper also provides a detailed analysis [1]. Here we provide results on **two additional Kaggle datasets**:
>
> | Method            | Avocado Prices | tabular-playground-series-jul-2021 |
> | ----------------- | -------------- | ---------------------------------- |
> | MLP               | 4.5141         | 2.0932                             |
> | MLP-Temporal      | 4.3536         | 2.1141                             |
> | MLP-Modulated     | **4.0379**     | **2.0294**                         |
> | MLP-PLR           | 5.0073         | 1.9825                             |
> | MLP-PLR-Temporal  | 5.1699         | 2.0638                             |
> | MLP-PLR-Modulated | **4.7394**     | **1.9684**                         |
> | TabM              | 4.3938         | 1.9521                             |
> | TabM-Temporal     | **3.6955**     | 1.9773                             |
> | TabM-Modulated    | 3.8976         | **1.8204**                         |
>
> Unlike the dataset in TabReD, **the temporal shift of these two datasets is not obvious**, so directly concatenating temporal embedding may not be effective. **Our method shows robustness**.
>
> > It would also be interesting to see whether it improves performance by similarly modulating based on the value of another continuous (or categorical) feature.
>
> This is a very good idea, and we've previously explored it, including combining it with ensemble methods. However, **current temporal embedding already incorporates many assumptions** (periodicity and trends). While features that satisfy a linear relationship or have a temporal period can be successfully used for modulation, other features with unknown periods are not. Furthermore, **selecting suitable features from among all available features is a key challenge**, as using unsuitable features for modulation can lead to missing input information and degraded performance. This is definitely a direction for future work, and it holds great potential in scenarios where the shift patterns can be formalized.
>
> > L258:  please provide some further details here.
>
> Thanks for pointing this out. Due to space constraints, we moved the details of this section to the appendix when we submitted the main paper. We will include more details here in the revision.
>
> > I find the colors of the points in Figure 4 quite difficult to see, especially due to the small size.
>
> Indeed, we will enlarge the sample points, highlight the different categories, and include a time axis in the revision to improve readability.
>
> We hope this response addresses your concern, and we will carefully address all the suggestions and correct any typos in the revision. Please feel free to raise any further questions or suggestions!
>
> ---
>
> [1] Rubachev, I., Kartashev, N., Gorishniy, Y., and Babenko, A. Tabred: A benchmark of tabular machine learning in-the-wild. In ICLR, 2025.

---

> ### Author Response · Authors · 2025-08-06
>
> Dear Reviewer,
>
> I hope this message finds you well. As the discussion period is nearing its end, with about three days remaining, I wanted to ensure that we have addressed all your concerns satisfactorily. If there are any additional points or feedback you would like us to consider, please let us know. Your insights are invaluable to us, and we are eager to address any remaining issues to improve our work.
>
> Thank you for your time and effort in reviewing our paper.

---

### Official Review · Reviewer_kbVu · 2025-07-02

**Clarity:** 3
**Significance:** 2
**Originality:** 3
**Rating:** 4
**Confidence:** 4

**Summary:**

The paper addresses the problem of temporal distribution shifts in tabular data by proposing a "feature-aware temporal modulation" method. This approach leverages a temporal embedding to dynamically adjust statistical properties (mean, standard deviation, skewness) of features, aiming to maintain semantic consistency over time. Empirical evaluations on the TabReD benchmark demonstrate improvements over both static and basic adaptive approaches.

**Questions:**

* How sensitive is the model’s performance to changes in the temporal embedding dimension? Could you elaborate on how you selected the embedding dimension used in your experiments?

* The focus seems to be limited on one type of distribution shifts. How does the approach perform under sudden, irregular, or non-smooth temporal shifts compared to gradual shifts?

* What is the standard deviation over different repetitions of the model? It would be very interesting to see

**Ethical Concerns:**

["NO or VERY MINOR ethics concerns only"]

**Final Justification:**

The paper presents a novel and well-motivated approach to handling temporal distribution shifts in tabular data via feature-aware temporal modulation. My initial concerns included the sensitivity to temporal embedding dimensionality, lack of significance analysis, and limited discussion of failure modes. The rebuttal addressed most of these points with new ablation studies, significance tests (Nemenyi), and clarifications on design choices and computational overhead. While the evaluation remains somewhat limited in dataset diversity and shift types, the conceptual contribution is valuable and well-supported by the provided evidence. I therefore maintain my borderline accept recommendation, as the strengths outweigh the remaining limitations.

**Limitations:**

* Evaluation on 8 datasets is very limited, especially as the rank was used as an aggregation statistic. Without per-dataset significance tests, no statistical claims about methods being superior to others are possible.

**Paper Formatting Concerns:**

-

**Quality:**

2

**Strengths And Weaknesses:**

Strengths:
The paper effectively identifies and illustrates the challenge of evolving semantics in temporal tabular data, which is a unique and novel perspective. Particularly, the objective vs. subjective semantics perspective is inspiring.

* Data-driven and problem-oriented analysis with insightful examples clearly illustrating the nature of distribution shifts and their impact.

* The introduction of feature-aware modulation seems like a conceptually innovative and intuitive idea for tabular data. (Although similar approaches already exist for time-series)

* Convincing evaluation on a well-designed benchmark (after the improvements of Cai et al.) with realistic prediction tasks


Weaknesses:
* The hyperparameter for the temporal embedding dimension is briefly mentioned but not thoroughly analyzed. An additional ablation study would strengthen the claims.

* Computational Efficiency: Although presented as a lightweight method, there is insufficient quantitative analysis regarding computational overhead.

* Limited discussion of scales of shift: Doesn’t yet characterize how severe temporal shifts must be before static models fail, leaving some uncertainty about practical thresholds.

* Some referenced adaptive methods (e.g., domain–adversarial, reweighting) are only briefly mentioned without analysis of their failure modes on temporal tasks.

* Limited analysis of the statistical significance of the approach. The authors use the mean rank over datasets to establish the superior performance of the proposed method. However,  on the datasets where the proposed approach ranks first, the difference to the next method always seems very small (Table 1). Given the missing analysis of significance per dataset, I consider it very likely that the proposed method is not truly better than the state-of-the-art. However, since the authors never claim that, I consider that a minor weakness. Even if it doesn’t improve performance, the approach is conceptually interesting and a good baseline for future work.

---

> ### Author Rebuttal · Authors · 2025-07-31
>
> We are grateful for your constructive suggestions! We will address your concerns in the following responses.
>
> > The hyperparameter for the temporal embedding dimension is briefly mentioned but not thoroughly analyzed. An additional ablation study would strengthen the claims.
>
> We further demonstrate the model performance achieved by **tuning parameters after fixing different temporal embedding dimensions**. Due to the limitation of hyperparameter space, all results are lower than those in Tab. 1.
>
> | Dimension | MLP-Temporal | MLP-Modulated |
> | --------- | ------------ | ------------- |
> | 8         | +0.20%       | **+0.65%**        |
> | 32        | +0.60%       | **+0.68%**        |
> | 128       | +0.92%       | **+1.47%**        |
> | 512       | +0.40%       | **+1.47%**        |
>
> We find that our **temporal modulation consistently outperforms temporal embedding**. Furthermore, there is an **inflection point** between the dimension and performance of temporal embedding, but not our temporal modulation. This demonstrates that our approach combines strong performance with scalability. By separating the conditional input (temporal) modality from the original feature modality, we reduce the impact of temporal input on the model backbone while enabling more refined temporal feature extraction, as also declared in response to reviewer dcxr.
>
> > How sensitive is the model’s performance to changes in the temporal embedding dimension? Could you elaborate on how you selected the embedding dimension used in your experiments?
>
> The temporal embedding dimension is indeed a very sensitive hyperparameter, as demonstrated in the experiments above. In our experiments, all hyperparameters were tuned using Optuna. The hyperparameter space for the temporal encoding dimension is int: [0, 8, 16, 32, 64, 128, 256, 512, 1024].
>
> > Doesn’t yet characterize how severe temporal shifts must be before static models fail, leaving some uncertainty about practical thresholds.
>
> This is a critical issue and difficult to quantify. However, in the design of temporal embedding, the hyperparameter search space already includes the option of not adding temporal embedding (i.e., dimension = 0), at which point the method degenerates to conventional methods. In other words, in our experiments, **whether to add temporal embedding is determined entirely through hyperparameter search**.
>
> You may also be interested in the response to the reviewer dcxr, which analyzes when to use our temporal modulation and when to use temporal embedding directly.
>
> > Some referenced adaptive methods (e.g., domain–adversarial, reweighting) are only briefly mentioned without analysis of their failure modes on temporal tasks.
>
> Domain-adversarial methods can only address domain-to-domain shifts and require partial target domain information, making them inappropriate for the temporal shift setting used in this paper. Reweighting methods can only address covariate and label shift, not concept shift, which is often severe in temporal shift datasets. Our temporal feature modulation aims to address concept shifts primarily caused by semantic evolution.
>
> > Limited analysis of the statistical significance of the approach.
>
> Due to the large dataset size in TabReD, there aren't enough experiments to support significant comparisons between all methods. Below are the Nemenyi tests for MLP and TabM:
>
> |               | MLP    | MLP-Temporal | MLP-Modulated |
> | ------------- | ------ | ------------ | ------------- |
> | MLP           | 1.0000 | 0.4237       | 0.0033        |
> | MLP-Temporal  | 0.4237 | 1.0000       | 0.1122        |
> | MLP-Modulated | 0.0033 | 0.1122       | 1.0000        |
>
> It can be seen that adding our modulation method to MLP significantly improves performance, while adding temporal embedding alone does not.
>
> |                | TabM   | TabM-Temporal | TabM-Modulated |
> | -------------- | ------ | ------------- | -------------- |
> | TabM           | 1.0000 | 0.7336        | 0.0631         |
> | TabM-Temporal  | 0.7336 | 1.0000        | 0.2909         |
> | TabM-Modulated | 0.0631 | 0.2909        | 1.0000         |
>
> Similar results are seen for TabM.
>
> In addition, our paper presents multiple evaluation metrics, including average rank (Tab. 1) and average percentage change (Fig. 5), providing a multi-perspective evaluation.
>
> > The focus seems to be limited to one type of distribution shifts. How does the approach perform under sudden, irregular, or non-smooth temporal shifts compared to gradual shifts?
>
> We acknowledge this limitation, but in the analysis in [1], the MMD heatmap shows that most real datasets have shifts in the form of trends and cycles. The data may indeed have unknown shifts in the test set, but for models deployed before this, there is no effective way to predict this.
>
> > Although presented as a lightweight method, there is insufficient quantitative analysis regarding computational overhead.
>
> We will introduce more detailed quantitative analysis in the revision. Our MLP with temporal modulation still significantly outperforms other models (such as the transformer) in terms of training efficiency and achieves better performance, which in part demonstrates the lightweight nature of our approach.
>
> > What is the standard deviation over different repetitions of the model? It would be very interesting to see.
>
> The following table provides the standard deviations of the random split results in the Appendix.
>
>
> | Method            | HI     | EO     | HD     | SH     | CT     | DE     | MR     | WE     |
> | ----------------- | ------ | ------ | ------ | ------ | ------ | ------ | ------ | ------ |
> | Linear            | 0.0009 | 0.0040 | 0.0018 | 0.0072 | 0.0003 | 0.0004 | 0.0073 | 0.0029 |
> | XGBoost           | 0.0003 | 0.0024 | 0.0091 | 0.0016 | 0.0002 | 0.0002 | 0.0001 | 0.0064 |
> | CatBoost          | 0.0008 | 0.0025 | 0.0022 | 0.0017 | 0.0003 | 0.0002 | 0.0001 | 0.0021 |
> | LightGBM          | 0.0005 | 0.0045 | 0.0081 | 0.0015 | 0.0003 | 0.0003 | 0.0004 | 0.0042 |
> | RandomForest      | 0.0002 | 0.0029 | 0.0017 | 0.0024 | 0.0001 | 0.0003 | 0.0000 | 0.0004 |
> | MLP               | 0.0042 | 0.0031 | 0.0011 | 0.0119 | 0.0004 | 0.0006 | 0.0001 | 0.0059 |
> | MLP-PLR           | 0.0014 | 0.0055 | 0.0102 | 0.0131 | 0.0003 | 0.0010 | 0.0002 | 0.0163 |
> | FT-T              | 0.0036 | 0.0095 | 0.0391 | 0.0082 | 0.0007 | 0.0014 | 0.0002 | 0.0084 |
> | TabR              | 0.0052 | 0.0055 | 0.0165 | 0.0068 | 0.0037 | 0.0033 | 0.0003 | 0.0067 |
> | ModernNCA         | 0.0007 | 0.0110 | 0.0074 | 0.0037 | 0.0005 | 0.0015 | 0.0004 | 0.0101 |
> | TabM              | 0.0014 | 0.0030 | 0.0128 | 0.0033 | 0.0007 | 0.0013 | 0.0002 | 0.0047 |
> | MLP-Modulated     | 0.0004 | 0.0028 | 0.0011 | 0.0068 | 0.0003 | 0.0012 | 0.0001 | 0.0052 |
> | MLP-PLR-Modulated | 0.0010 | 0.0041 | 0.0082 | 0.0027 | 0.0003 | 0.0010 | 0.0001 | 0.0078 |
> | TabM-Modulated    | 0.0008 | 0.0020 | 0.0090 | 0.0034 | 0.0006 | 0.0012 | 0.0002 | 0.0073 |
>
> We hope this response addresses your concern. Please feel free to raise any further questions or suggestions!
>
> ---
>
> [1] Cai, H.-R. and Ye, H.-J. Understanding the limits of deep tabular methods with temporal shift. CoRR, abs/2502.20260, 2025.

---

> ### Author Response · Authors · 2025-08-06
>
> Dear Reviewer,
>
> I hope this message finds you well. As the discussion period is nearing its end, with about three days remaining, I wanted to ensure that we have addressed all your concerns satisfactorily. If there are any additional points or feedback you would like us to consider, please let us know. Your insights are invaluable to us, and we are eager to address any remaining issues to improve our work.
>
> Thank you for your time and effort in reviewing our paper.

---

> > ### Comment · Reviewer_kbVu · 2025-08-07
> >
> > Thank you for the detailed and thoughtful rebuttal. I appreciate the additional empirical results, particularly the ablation on temporal embedding dimension and the significance tests. These additions strengthen the paper and address most of my initial concerns. While I still believe that broader evaluation and more discussion of failure modes would further improve the work, I agree that the proposed method is conceptually interesting and a promising step toward handling temporal shifts in tabular data. I maintain my score and recommend acceptance.

---

> > > ### Author Response · Authors · 2025-08-07
> > >
> > > We deeply appreciate your constructive suggestions and valuable feedback throughout the review process, which have greatly helped us improve our work! The changes addressed during the rebuttal will be carefully reflected in our revised version. Thank you again for your time and engagement!

---

### Official Review · Reviewer_Gi1z · 2025-07-03

**Clarity:** 4
**Significance:** 3
**Originality:** 4
**Rating:** 5
**Confidence:** 4

**Summary:**

The paper tackles a problem of temporal shift in tabular data. It first identifies that evolving feature semantics (the difference between relative and absolute feature values) is tied to temporal shift in tabular data (via visualization of feature distributions across time, hypothesis and examples). Authors then propose a dynamic adaptation method which is a feature/neuron/logit modulation based on timestamp -- which helps normalize features and model internals across timestamps. The authors show significant gains on a real world tabular benchmark TabReD.

**Questions:**

-   Could you provide results for the default TabReD splits, and discuss those together with results on random splits? It is interesting how the proposed technique interacts with different validation and splitting strategies (for example, why are there gains in random splits too, where there is presumably less shift).
-   What encourages the modulator behaviour illustrated in Figure 4? A deeper investigation in mechanisms behind this effect would be very insightful.

**Ethical Concerns:**

["NO or VERY MINOR ethics concerns only"]

**Final Justification:**

I've read all the other reviews and author responses. I belive the submission clearly meets the bar for acceptance.

I also want to address the concern of reviewer `dcxr` regarding small 1-2% gains on real-world datasets – those gains are actually substantial when it comes to this particular benchmark, furthermore seing *more or less* consistent gains over strong baselines such as MLP-PLR or TabM is also an encouraging signal regarding the methods applicability.

**Limitations:**

Limitations are adequately discussed

**Paper Formatting Concerns:**

Paper is typeset well, there are no issues with the formatting.

**Quality:**

4

**Strengths And Weaknesses:**

**Strengths**:

-   The authors find a compelling explanation on how temporal shift in tabular data happens and use it to address it in a clear and simple way via feature modulation.
-   The empirical results seem strong (modulo notes discussed in the weaknesses section below).
-   The writing is clear, engaging and easy to understand.

**Weaknesses**:

-   The authors use a validation setup from the follow-up paper, which may be less standard than the one proposed in the original benchmark. In addition to including results on random splits, it would be great if authors provide results on the original splits from the TabReD benchmark.
-   Writing is not precise enough on how the modulator is trained &#x2013; I assume that it's just trained as ususal, this leads to the next question (*most important in this review for me* - addressing it decently would lead to score increase with high probability)
-   What precisely encourages such "normalizing" behaviour (well illustrated on synthetic data in Figure 4). Is it the training procedure plus the hold-out validation on a shifted subset? Is it some kind-of optimization dynamic specific to the modulator architecture? A deeper investigation beyond simple high level ablation would be very meaningful and make paper much stronger.

---

> ### Author Rebuttal · Authors · 2025-07-31
>
> We appreciate your thoughtful comments! We will address your concerns in the following responses.
>
> > In addition to including results on random splits, it would be great if authors provide results on the original splits from the TabReD benchmark.
>
> We further show the model performance before and after adding our temporal modulation to the **original temporal split in TabReD** [1].
>
> | Method            | HI         | EO         | HD         | SH         | CT         | DE         | MR         | WE         |
> | ----------------- | ---------- | ---------- | ---------- | ---------- | ---------- | ---------- | ---------- | ---------- |
> | Linear            | 0.9388     | 0.5731     | 0.8174     | 0.2560     | 0.4879     | 0.5587     | 0.1744     | 1.7465     |
> | XGBoost           | 0.9609     | 0.5764     | **0.8627** | 0.2475     | 0.4823     | **0.5459** | 0.1616     | **1.4699** |
> | CatBoost          | 0.9612     | 0.5671     | 0.8588     | 0.2469     | 0.4824     | 0.5464     | 0.1619     | 1.4715     |
> | LightGBM          | 0.9600     | 0.5633     | 0.8580     | 0.2452     | 0.4826     | 0.5474     | 0.1618     | 1.4723     |
> | RandomForest      | 0.9537     | 0.5755     | 0.7971     | 0.2623     | 0.4870     | 0.5565     | 0.1653     | 1.5839     |
> | MLP               | 0.9404     | 0.5866     | 0.4730     | 0.2802     | 0.4820     | 0.5526     | 0.1624     | 1.5331     |
> | MLP-PLR           | 0.9592     | 0.5816     | 0.8448     | 0.2412     | 0.4811     | 0.5533     | 0.1616     | 1.5185     |
> | FT-T              | 0.9562     | 0.5791     | 0.5301     | 0.2600     | 0.4814     | 0.5534     | 0.1627     | 1.5155     |
> | TabR              | 0.9527     | 0.5727     | 0.8442     | 0.2676     | 0.4818     | 0.5557     | 0.1625     | 1.4782     |
> | ModernNCA         | 0.9571     | 0.5712     | 0.8487     | 0.2526     | 0.4817     | 0.5523     | 0.1631     | 1.4977     |
> | TabM              | 0.9590     | 0.5952     | 0.8549     | 0.2465     | 0.4799     | 0.5522     | **0.1610** | 1.4852     |
> | MLP-Modulated     | 0.9538     | 0.5832     | 0.4753     | 0.2966     | 0.4811     | 0.5587     | 0.1623     | 1.5157     |
> | MLP-PLR-Modulated | 0.9576     | 0.5824     | 0.8434     | **0.2353** | 0.4805     | 0.5605     | 0.1616     | 1.5188     |
> | TabM-Modulated    | **0.9634** | **0.5978** | $--$       | 0.2415     | **0.4796** | 0.5533     | 0.1614     | 1.4813     |
>
> The results are **consistent** with those in paper, and our method still achieves strong performance on the original temporal split of TabReD. This shows the **robustness** of our method in different validation strategies.
>
> > It is interesting how the proposed technique interacts with different validation and splitting strategies.
>
> In our understanding, the key to resolving temporal shift lies in achieving **distributional and temporal extrapolation capabilities** (**L37-L47**). The difference is that the training protocol in [2] modifies the validation split to allow the model's performance to be better extrapolated to test time (so does the random split), which is the foundation for model learning. Our temporal modulation (or directly concatenate temporal embedding) finds a solution to the extrapolation capability within a given split. We believe that **the effects of these two approaches are orthogonal**, affecting the final model performance in different ways. This also explains why our method has similar effects on all validation splits.
>
> > Writing is not precise enough on how the modulator is trained – I assume that it's just trained as usual.
>
> Yes, the modulator is trained **end-to-end along with the model backbone**.
>
> > What precisely encourages such "normalizing" behaviour?
>
> Comparing this phenomenon with normalization is quite instructive. Although we haven't discussed this before, we believe that our modulation method can **indeed be considered a special form of normalization**, but a learnable and more general form because it supports not only scale changes but also skew changes. From this perspective, we believe that this is not the result of hold-out validation (as just mentioned, these two approaches are orthogonal in our understanding), but is more similar to the **RevIN method [3] in time series prediction**: in time series prediction, by learning a scaling and offset coefficient for each subsequence, the distribution shift of the time series can be effectively addressed. **Corresponding to our method**, we first use the temporal embedding to guide the model "which samples belong to the same subsequence and should be scaled together" in temporal tabular data, and then learn the coefficients of each "subsequence" in an end-to-end form, thereby learning a **transformation for each specific temporal stage**.
>
> We hope this response addresses your concern. We're open to further discussion and would appreciate any additional thoughts you may have!
>
> ---
>
> [1] Rubachev, I., Kartashev, N., Gorishniy, Y., and Babenko, A. Tabred: A benchmark of tabular machine learning in-the-wild. In ICLR, 2025.
>
> [2] Cai, H.-R. and Ye, H.-J. Understanding the limits of deep tabular methods with temporal shift. CoRR, abs/2502.20260, 2025.
>
> [3] Kim, T., Kim, J., Tae, Y., Park, C., Choi, J.-H., and Choo, J. Reversible Instance Normalization for Accurate Time-Series Forecasting against Distribution Shift. In ICLR, 2022.

---

> ### Author Response · Authors · 2025-08-06
>
> Dear Reviewer,
>
> I hope this message finds you well. As the discussion period is nearing its end, with about three days remaining, I wanted to ensure that we have addressed all your concerns satisfactorily. If there are any additional points or feedback you would like us to consider, please let us know. Your insights are invaluable to us, and we are eager to address any remaining issues to improve our work.
>
> Thank you for your time and effort in reviewing our paper.

---

> > ### Comment · Reviewer_Gi1z · 2025-08-07
> >
> > Thanks for the extensive rebuttal. I have no more concerns, the clarification and analogy to the RevIN method from time-series helps, thanks.

---

> > > ### Author Response · Authors · 2025-08-09
> > >
> > > We deeply appreciate your insightful comments and valuable feedback throughout the review process. The changes addressed during the rebuttal will be carefully reflected in our revised version. Thank you again for your time and engagement!

---

> > > ### Author Response · Authors · 2025-08-09
> > >
> > > Dear Reviewer,
> > >
> > > We sincerely appreciate your mandatory acknowledgement. If we understand correctly, this implies that our rebuttal and follow-up responses have adequately addressed your concerns. We would be truly grateful if you might consider reflecting this in your score and supporting the acceptance of our paper.
> > > Thank you once again for your valuable feedback and for taking the time to review our work.

---

### Official Review · Reviewer_dcxr · 2025-07-03

**Clarity:** 3
**Significance:** 2
**Originality:** 2
**Rating:** 3
**Confidence:** 4

**Summary:**

The paper aims to tackle the problem of temporal distribution shift in tabular data. The authors assert that distribution shifts can arise because the semantics of individual features (e.g., high income) can change as the global feature distribution drifts. To mitigate this problem, the authors propose Feature-Aware Temporal Modulation: a lightweight hypernetwork that, given a temporal embedding, generates the parameters of a lightweight linear and nonlinear transform that we can insert into multiple parts of a neural network to mitigate temporal drift. Experiments on the TabReD benchmark show that the proposed method achieves the best average rank among the experimented baselines.

**Questions:**

Please refer to the Weaknesses section.

**Ethical Concerns:**

["NO or VERY MINOR ethics concerns only"]

**Final Justification:**

During the review and discussion period, my raised issues and current thoughts are:

**[Point 1]** The empirical gains are very small (1~2%). Against the competing baseline(temporal embedding[1]), the performance is very neck-and-neck, with the competing baseline outperforming the proposed method on multiple datasets. This is even more so when considering the variance of the experiments.

Since the gains on real-world datasets are marginal, I have requested measurements against a synthetic dataset, which resulted in relatively more gains than real datasets. However, the gains are attributed to other types of shifts, while the main focus of writing in the manuscript was concept shift. To this end, I think that integrating this result would be challenging without a major revision of the manuscript.


**[Point 2]** To the best of my understanding, the paper tries to implicitly claim the contribution credit for raising the importance of the temporal distribution shift problem in tabular data. However, this contribution is attributed to a prior work[1] and not this manuscript. Although the authors do cite [1], the writing and visual analyses of the manuscript attenuate or avoid the presence of the prior work. For instance, the paper does not clarify the boundary between settings where the proposed method helps and where just supplying a temporal embedding[1] to the model and “letting the network figure it out” could be better.

To this end, I have requested active comparisons between the proposed method and [1] - e.g., to compare the evolution of decision boundaries over time (Figure 4 of the manuscript) so that we can observe whether [1] fails to capture the evolution while the proposed method succeeds. However, the authors say that *"Similar to our method, temporal embedding allows the model to change its decision boundary over time."* which entails that [1] can also capture the evolution of the decision boundary. Moreover, incorporating this result into the manuscript would require a major revision of the narrative and visualizations.


Considering the points above, I wish to keep my current rating while lowering the originality score.

[1] Cai, H.-R. and Ye, H.-J. Understanding the limits of deep tabular methods with temporal shift. CoRR, abs/2502.20260, 2025.

**Limitations:**

yes

**Paper Formatting Concerns:**

I do not notice any major formatting issues.

**Quality:**

3

**Strengths And Weaknesses:**

**Strengths**

- The work tackles temporal distribution shift in tabular data, which is an important but under-recognized problem.
- The modulator is very lightweight - it has a small parameter count.

**Weaknesses**

- The reported improvements on TabReD are small (1–2 % points). To further assess the method’s distinctive strengths, I believe that the authors could construct and experiment on more controlled synthetic datasets that can show the strengths of the proposed method. For example (these are examples, not definitive nor exhaustive):
    - Experiment on synthetic covariate-shift, label-shift, and concept-shift scenarios.
    - Observe how latent representations drift over time under each shift scenario.
    - Compare the latent space of the proposed method compared to supplying vanilla temporal embeddings to the baseline model.
- The paper does not clarify the boundary between settings where the proposed method helps and where just supplying a temporal embedding to the model and “letting the network figure it out” could be better:
    - Distribution shape change (e.g., what if a feature that is initially unimodal becomes bimodal)
    - High dimensionality of input feature space
    - Categoricals or ordinal features

---

> ### Author Rebuttal · Authors · 2025-07-31
>
> Thank you for your detailed and constructive review! We will address your concerns in the following responses.
>
> > The reported improvements on TabReD are small (1–2%). To further assess the method’s distinctive strengths, I believe that the authors could construct and experiment on more controlled synthetic datasets that can show the strengths of the proposed method.
>
> We are very grateful for your valuable suggestions and have supplemented some further analysis:
>
> > > Experiment on synthetic covariate-shift, label-shift, and concept-shift scenarios.
>
> The **Fig. 4** in paper has already demonstrated the concept shift. We further constructed the above three different shifts based on a same dataset and performed numerical comparisons.
>
> | Shift | MLP        | MLP-Temporal | MLP-Modulated |
> | -------------- | ---------- | ------------ | ------------- |
> | None           | **0.9940** | 0.9758       | 0.9928        |
> | Concept        | 0.8850     | 0.9833       | **0.9878**    |
> | Label          | 0.9216     | 0.9176       | **0.9455**    |
> | Covariant      | 0.9045     | 0.9039       | **0.9146**    |
>
> The above results show that both temporal embedding and temporal modulation **focus on addressing concept shift**, which is consistent with our expectations and the focus of the paper. However, for covariate shift and label shift (which we obtained by sampling different time slices of the dataset), we found that **simply adding temporal embeddings resulted in performance degradation, while our temporal modulation improved performance**. We believe this may be because the model's response to covariate shift and label shift primarily depends on its inductive bias, while temporal modulation directly acts on the decision boundary and always attempts to **correct any potential inductive bias**.
>
> This also explains why our method is better than simply adding temporal embedding: the temporal shift in real data is often a complex superposition of the three types of shift, and our method achieves better performance on all three shifts.
>
> These results are encouraging and we will discuss them further in the revision. In addition, we will also include **visualization results** in the revision.
>
> > > Observe how latent representations drift over time under each shift scenario.
>
> If we analyze the overall distribution of the latent representation, we believe this may be similar to the conclusion of the MMD heatmap in [1], that is, the model representation $\phi(x)$ is nearly **stable when no temporal information is used**, but the target mapping $f$ is constantly changing. Therefore, we should pay more attention to the changes in the mapping and how the modulation is corrected, rather than how the model representation itself changes. If we try to analyze it on feature level, since each dimension of the model representation is often a fusion of multiple features, it is actually difficult to know whether the type of shift in the concept of each feature is as shown in **Fig. 1**, or whether there is no shift at all. If you have better ideas or suggestions, please feel free to raise them!
>
> > > Compare the latent space of the proposed method compared to supplying vanilla temporal embeddings to the baseline model.
>
> This is also an interesting research direction, and we will include visualizations in the revision if possible. In our understanding, the fundamental difference between temporal modulation and simply adding temporal embedding is that **temporal modulation isolates temporal information from the modality of the conditional input, rather than conflating it with the original input modality**. This prevents the model input from being **affected** by overly complex temporal embedding and enables more **refined** temporal feature extraction. (For example, when simply adding temporal embedding, the embedding dimension tuned by Optuna often does not exceed 100, as the model's input feature dimension is also only ~100. However, in our temporal modulation, although we set an upper limit of 1024 for the embedding dimension, many experiments still reach this limit.) In contrast to latent representation, we believe that simply adding temporal embedding can lead to a serious **curse of dimensionality**, while temporal modulation **fundamentally avoids this problem**.
>
> Besides, we believe that a 1-2% performance improvement is not small. Without incorporating any temporal information, the performance gap between the most basic MLP model and the strongest tested TabM model is approximately 2.7%. All deep methods except TabM and MNCA have a performance gap of less than 2% compared to MLP. This makes the dynamic MLP with our temporal modulation outperform all static deep methods except TabM and MNCA under temporal shifts. This can also be seen in **Tab. 1**.
>
> > The paper does not clarify the boundary between settings where the proposed method helps and where just supplying a temporal embedding could be better.
>
> We acknowledge that this is an important question, one that the paper does not explore further. This may require more detailed experiments. Here we only list a few possible observations:
>
> 1. For **ordered but discrete numerical features** (e.g., int-type numerical features), temporal modulation does not necessarily yield better results than temporal embedding. This may be because the knowledge learned by the model is already discrete, and directly adding modulation to it makes it difficult to optimize, resulting in worse results than adding temporal embedding to the extra dimension.
> 2. Compared to **categorical features**, we did not observe the results in 1. This may be because we use one-hot encoding for categorical features, allowing our modulation to apply independently to each category. This provides greater flexibility.
> 3. For **features with strong non-normality** (e.g., long-tailed distributions), our temporal modulation yields greater improvements. This may be because our modulation includes a correction for skewness, which allows the model to learn better.
> 4. We rarely observed distribution changes such as "unimodal to bimodal" (in other words, we generally found corresponding relationships with different peaks, although the density differences may be large), but we believe this may be a difficult issue. Our time modulation may only be able to achieve this correction by transforming all distributions into more normal distributions.
> 5. **Input dimensionality is indeed a key issue**. As mentioned in the previous question, our time modulation fundamentally addresses the curse of dimensionality that can result from input temporal embedding. **You may be interested in the response to reviewer kbVu**, where we analyze the performance of temporal embedding and our temporal modulation **when different time dimensions are fixed**. The results show that our method achieves both performance and scalability. This means that temporal modulation will perform better in scenarios with more complex temporal features.
>
> We hope this response addresses your concern. Thank you again for your constructive feedback, which will help us improve our work! Please feel free to raise any further questions or suggestions!
>
> ---
>
> [1] Cai, H.-R. and Ye, H.-J. Understanding the limits of deep tabular methods with temporal shift. CoRR, abs/2502.20260, 2025.

---

> > ### Comment · Reviewer_dcxr · 2025-08-06
> >
> > > Besides, we believe that a 1-2% performance improvement is not small.
> >
> > What are the standard error values of the experiments? How many times are the experiments repeated?

---

> > > ### Author Response · Authors · 2025-08-06
> > >
> > > Thank you for your reply!
> > >
> > > The standard deviations of our experimental results are reported in **Appendix E**. All results are averaged over **15 runs** with different random seeds, as described in **Appendix B**. This is a common practice in tabular learning, as adopted in prior works [1–5] and TabReD [6].
> > >
> > > If you have any further concerns or questions, please feel free to let us know!
> > >
> > > ---
> > >
> > > [1] Gorishniy, Y., Rubachev, I., Khrulkov, V., and Babenko, A. Revisiting deep learning models for tabular data. In NeurIPS, pp. 18932–18943, 2021.
> > >
> > > [2] Gorishniy, Y., Rubachev, I., and Babenko, A. On embeddings for numerical features in tabular deep learning. In NeurIPS, pp. 24991–25004, 2022.
> > >
> > > [3] Gorishniy, Y., Rubachev, I., Kartashev, N., Shlenskii, D., Kotelnikov, A., and Babenko, A. Tabr: Tabular deep learning meets nearest neighbors. In ICLR, 2024.
> > >
> > > [4] Gorishniy, Y., Kotelnikov, A., and Babenko, A. Tabm: Advancing tabular deep learning with parameter-efficient ensembling. In ICLR, 2025.
> > >
> > > [5] Ye, H.-J., Yin, H.-H., Zhan, D.-C., and Chao, W.-L. Revisiting nearest neighbor for tabular data: A deep tabular baseline two decades later. In ICLR, 2025.
> > >
> > > [6] Rubachev, I., Kartashev, N., Gorishniy, Y., and Babenko, A. Tabred: A benchmark of tabular machine learning in-the-wild. In ICLR, 2025.

---

> > > > ### Comment · Reviewer_dcxr · 2025-08-07
> > > >
> > > > The authors claim that a 1~2% increase is not small, but against the competing baseline(temporal embedding[1]), the performance is very neck-and-neck, with the competing baseline outperforming the proposed method on multiple datasets. This is even more so when considering the variance of the experiments.
> > > >
> > > > As the contribution credit for raising the importance of the temporal distribution shift problem in tabular data is attributed to a prior work[1] and not this manuscript, and since that work already proposed a simple and effective baseline solution(temporal embedding), the main focus of comparison should be between [1] and the proposed method.
> > > >
> > > > Thus, it is also crucial to supply the results for [1] in Figure 4 to observe the relative improvement against [1]. I am aware that graphical components cannot be attached currently, but nevertheless, I believe that such are necessary to assess the relative improvement of the proposed method against competing baselines, not the vanilla baseline.
> > > >
> > > >
> > > > [1] Cai, H.-R. and Ye, H.-J. Understanding the limits of deep tabular methods with temporal shift. CoRR, abs/2502.20260, 2025.

---

> > > > > ### Author Response · Authors · 2025-08-08
> > > > >
> > > > > Thank you for your insightful comment. We appreciate your close reading and your suggestion to more directly compare our method to the temporal embedding baseline proposed in [1], which we acknowledge as a strong and effective approach to handling temporal distribution shifts in tabular data.
> > > > >
> > > > > We fully agree that the primary point of comparison should be between our proposed *temporal modulation* and the *temporal embedding* baseline, and we have accordingly extended our experiments to address this more thoroughly. We have now included the results of the temporal embedding baseline in the setting of Figure 4, as well as in several other comparative evaluations. While we are unable to include additional figures in this rebuttal, we will incorporate these updated comparisons in the revised version and summarize our key observations:
> > > > >
> > > > > 1. **Temporal embedding introduces dynamic decision boundaries, but with higher complexity.**
> > > > >    Similar to our method, temporal embedding allows the model to change its decision boundary over time. However, we find that this added flexibility can results in overfitting, leading to more misclassification.
> > > > > 2. **Dimensionality expansion introduces instability in learned representations.**
> > > > >    Because temporal embeddings are directly concatenated to the input features, the input space expands significantly, especially when the embedding dimension increases. As a result, latent representations derived from such inputs no longer maintain a clear correspondence to the original features, complicating interpretability and degrading stability.
> > > > > 3. **Curse of dimensionality under large temporal embeddings.**
> > > > >    We observe that increasing the embedding dimension causes the latent space of temporal embedding models to become more dispersed (as revealed via PCA visualizations), which we interpret as a manifestation of the curse of dimensionality. In contrast, our modulation approach operates on the model parameters rather than the input space and thus avoids this issue.
> > > > >
> > > > > It is worth noting that Figure 4 essentially constructs a concept shift dataset. In our previous rebuttal, we have also provided numerical comparisons between temporal embedding and temporal modulation under covariate/label/concept shift.
> > > > >
> > > > > **Additional comparisons** include:
> > > > >
> > > > > - **Fixed embedding dimension experiments** (see our response to Reviewer kbVu), where we demonstrate that performance of temporal embedding plateaus or even drops as the embedding dimension increases, while our method continues to improve or remains stable.
> > > > > - **Significance tests** (also in response to Reviewer kbVu), which show that our temporal modulation yields statistically significant improvements over static baselines, whereas temporal embedding often does not.
> > > > >
> > > > > While both temporal embedding and temporal modulation aim to address the same underlying problem, we emphasize that they do so **from fundamentally different perspectives**. Since the goal of handling temporal shift is to enable models to adapt over time, temporal embedding introduces ***input-level* dynamics** (dynamic data), while our temporal modulation introduces ***parameter-level* dynamics** (dynamic mapping), which truly achieves a dynamic model.
> > > > >
> > > > > Finally, **we would like to reiterate that the primary contribution of our work lies not merely in empirical gains, but in proposing a novel and lightweight modulation mechanism that enables temporal adaptation via dynamic model**. As temporal shift research in tabular learning is still in its early stages, we hope this work contributes to broadening the set of tools and perspectives available for tackling this important but underexplored challenge.
> > > > >
> > > > > We thank you again for your thoughtful feedback. If you have any additional suggestions, we would be happy to address them further.

---

> > > > > ### Author Response · Authors · 2025-08-09
> > > > >
> > > > > Dear Reviewer,
> > > > >
> > > > > We greatly appreciate your valuable feedback. With less than 10 hours remaining in the rebuttal phase, we sincerely hope our previous responses have addressed your concerns. If you have any further questions, we would be more than happy to clarify them within the remaining time. We look forward to hearing from you.

---

> ### Author Response · Authors · 2025-08-06
>
> Dear Reviewer,
>
> I hope this message finds you well. As the discussion period is nearing its end, with about three days remaining, I wanted to ensure that we have addressed all your concerns satisfactorily. If there are any additional points or feedback you would like us to consider, please let us know. Your insights are invaluable to us, and we are eager to address any remaining issues to improve our work.
>
> Thank you for your time and effort in reviewing our paper.

---

### Decision · Program_Chairs · 2025-09-17

**Decision:**

Accept (poster)

**Comment:**

The paper introduces a feature-aware modulation mechanism to handle temporal distribution shifts in tabular data by aligning feature semantics over time through lightweight modulation of distributional statistics such as scale, skewness, and bias. This approach enables models to better balance generalization and adaptability compared to static models or simple temporal embeddings, and empirical evaluation on the TabReD benchmark demonstrates consistent, if modest, improvements. To further strengthen the work, the authors are encouraged to candidly discuss its limitations, particularly around the magnitude of empirical improvements, the boundaries of when modulation is superior to temporal embedding, and potential weaknesses in high-dimensional or categorical feature settings. A more open reflection on these limitations would enhance the credibility of the contribution and help guide future research on temporal shift in tabular learning.